# STAGE-WISE DYNAMICS OF CLASSIFIER-FREE GUIDANCE IN DIFFUSION MODELS

**Cheng Jin, Qitan Shi & Yuantao Gu***
Department of Electronic Engineering, Tsinghua University
Beijing, China
{jinc21,sqt24}@mails.tsinghua.edu.cn, gyt@tsinghua.edu.cn

## ABSTRACT

Classifier-Free Guidance (CFG) is widely used to improve conditional fidelity in diffusion models, but its impact on sampling dynamics remains poorly understood. Prior studies, often restricted to unimodal conditional distributions or simplified cases, provide only a partial picture. We analyze CFG under multimodal conditionals and show that the sampling process unfolds in three successive stages. In the Direction Shift stage, guidance accelerates movement toward the weighted mean, introducing initialization bias and norm growth. In the Mode Separation stage, local dynamics remain largely neutral, but the inherited bias suppresses weaker modes, reducing global diversity. In the Concentration stage, guidance amplifies within-mode contraction, diminishing fine-grained variability. This unified view explains a widely observed phenomenon: stronger guidance improves semantic alignment but inevitably reduces diversity. Experiments support these predictions, showing that early strong guidance erodes global diversity, while late strong guidance suppresses fine-grained variation. Moreover, our theory naturally suggests a time-varying guidance schedule, and empirical results confirm that it consistently improves both quality and diversity.

## 1 INTRODUCTION

Diffusion models have driven remarkable progress in generative modeling, achieving state-of-the-art results across images, video, audio, and multimodal domains (Rombach et al., 2022; Podell et al., 2023; Cao et al., 2024; Zhang et al., 2025). A central requirement in practice is conditional generation, where outputs must follow prompts, class labels, or other structured signals. Among existing strategies, *Classifier-Free Guidance* (CFG) (Ho & Salimans, 2021) has become the de facto standard due to its simplicity and effectiveness, powering nearly all large-scale diffusion pipelines. Numerous variants further extend its influence (Jin et al., 2025; Gao et al., 2025; Malarz et al., 2025; Chung et al., 2025; Sadat et al., 2025; Castillo et al., 2025). Yet despite its widespread adoption, the theoretical underpinnings of CFG remain poorly understood.

While recent work has advanced the theory of diffusion sampling (Benton et al., 2023; Chen et al., 2023; Cai & Li, 2025; Li et al., 2025a), the analysis of CFG itself is still in its infancy, hindered by its heuristic formulation. Prior studies fall into two categories. The first assumes a unimodal conditional distribution—typically a single Gaussian—yielding clean derivations but overlooking the multimodal nature of real-world tasks (Chidambaram et al., 2024; Wu et al., 2024; Bradley & Nakkiran, 2024; Xia et al., 2024; Jin et al., 2025; Pavasovic et al., 2025; Li et al., 2025b). The second relaxes these assumptions but imposes only weak conditions, producing broad qualitative insights without sharp predictions (Li & Jiao, 2025). Consequently, several well-documented empirical phenomena remain theoretically elusive, most notably the collapse of diversity under large guidance weights (Ho & Salimans, 2021). Explaining this diversity loss is both a key theoretical challenge and a practical necessity for improving conditional generation. A detailed discussion of these related works can be found in Appendix.

In this work, we move beyond the restrictive unimodality assumption by modeling conditional distributions as Gaussian mixtures. This perspective reveals that guided sampling follows a natural

---

*Corresponding author

*three-stage structure*. In the *Direction Shift* stage (early, high-noise regime), trajectories are drawn toward the class-weighted mean, and guidance amplifies this attraction, inducing initialization bias and norm inflation. In the *Mode Separation* stage (intermediate regime), the dynamics remain locally neutral, but the inherited bias suppresses weaker modes, reducing global diversity. In the *Concentration* stage (late regime), contraction within modes is intensified by strong guidance, suppressing local variability and fine-grained diversity.

We empirically validate these predictions on state-of-the-art diffusion models rather than toy settings. To test our framework, we first compare early-high, late-high, and all-high schedules, confirming that strong early guidance reduces global diversity. To probe the late-stage behavior, we initialize trajectories from identical noise and introduce small perturbations during the intermediate stage. We observe that excessively large late-stage guidance forces trajectories to converge toward nearly identical fine details, thereby diminishing local diversity. Beyond these validation studies, we further conduct an additional experiment with a time-varying guidance schedule which can improve performance. Although this is not the central theoretical contribution of this work, it underscores the practical relevance of our analysis. Our implementation is publicly available at `https://github.com/sqt24/tvcfg`.

**Contributions.** This work makes three contributions: (i) We provide the first theoretical framework for analyzing CFG under multimodal conditional distributions, revealing a natural three-stage structure of the sampling dynamics. (ii) We empirically validate the central predictions of this framework: strong early guidance reduces global diversity, while strong late guidance suppresses local variability. (iii) As a byproduct of our analysis, we observe that varying guidance strength over time can improve the quality–diversity trade-off. Together, these results unify theoretical and empirical perspectives and establish a foundation for the principled design of guided diffusion samplers.

## 2 PRELIMINARY

### 2.1 DIFFUSION MODEL

Diffusion models generate data by inverting a forward noising process that progressively transforms samples from the target distribution $p_{\text{data}}(\boldsymbol{x})$ into a simple Gaussian reference. The forward corruption can be written as

$$\mathrm{d}\boldsymbol{x}_t = -\alpha(t)\boldsymbol{x}_t\mathrm{d}t + \sqrt{2\beta(t)}\mathrm{d}\boldsymbol{w}_t, \quad \boldsymbol{x}_0 \sim p_{\text{data}}, \quad t \in [0, T], \tag{1}$$

where $\alpha(t)$ and $\beta(t)$ control the drift and diffusion, and $\boldsymbol{w}_t$ is a Wiener process. As $t$ grows, the distribution $p_t$ converges toward an isotropic Gaussian, ensuring that sampling can begin from a tractable prior.

By time-reversal theory (Anderson, 1982), sampling is performed through the reverse SDE

$$\mathrm{d}\boldsymbol{x}_t = \big[ -\alpha(t)\boldsymbol{x}_t - 2\beta(t)\nabla_{\boldsymbol{x}_t} \log p_t(\boldsymbol{x}_t)\big]\mathrm{d}t + \sqrt{2\beta(t)}\mathrm{d}\bar{\boldsymbol{w}}_t, \tag{2}$$

where $\nabla_{\boldsymbol{x}_t} \log p_t(\boldsymbol{x}_t)$ is the *score function*, representing the gradient of the log-density, and $\bar{\boldsymbol{w}}_t$ is a backward Wiener process.

Modern implementations typically adopt the *probability flow ODE* (Song et al., 2021), which eliminates randomness while preserving the same marginal distributions:

$$\frac{\mathrm{d}\boldsymbol{x}_t}{\mathrm{d}t} = -\alpha(t)\boldsymbol{x}_t - \beta(t)\nabla_{\boldsymbol{x}_t} \log p_t(\boldsymbol{x}_t). \tag{3}$$

Its deterministic nature allows the use of high-order ODE solvers (Lu et al., 2022; Zhao et al., 2023), enabling efficient generation with few function evaluations and making theoretical analysis more tractable. This ODE view is particularly important for understanding how guidance mechanisms reshape the underlying dynamics.

**Conditional generation.** When conditioning on external information $y$ (e.g., a class label or a text prompt), one simply replaces the unconditional score with the conditional score:

$$\frac{\mathrm{d}\boldsymbol{x}_t}{\mathrm{d}t} = -\alpha(t)\boldsymbol{x}_t - \beta(t)\nabla_{\boldsymbol{x}_t} \log p_t(\boldsymbol{x}_t \mid y). \tag{4}$$

This *conditional probability flow ODE* produces samples consistent with $y$ while retaining the computational benefits of the ODE formulation. However, in practice the learned conditional score may be weak or under-confident, leading to poor semantic alignment with the conditioning signal.

## 2.2 CLASSIFIER-FREE GUIDANCE

While the conditional ODE equation 4 incorporates side information $y$, its influence in practice is often insufficient. *Classifier-Free Guidance* (CFG) addresses this issue by extrapolating between the unconditional score $\nabla_{\boldsymbol{x}_t} \log p_t(\boldsymbol{x}_t)$ and the conditional score $\nabla_{\boldsymbol{x}_t} \log p_t(\boldsymbol{x}_t \mid y)$ predicted by the same model:

$$\hat{\boldsymbol{s}}_t(\boldsymbol{x}_t; y, \omega) = (1 - \omega)\, \nabla_{\boldsymbol{x}_t} \log p_t(\boldsymbol{x}_t) + \omega\, \nabla_{\boldsymbol{x}_t} \log p_t(\boldsymbol{x}_t \mid y), \quad \omega > 1. \tag{5}$$

Here the *guidance scale* $\omega$ controls the trade-off: $\omega = 1$ reduces to the plain conditional model, whereas larger values enforce stronger alignment with $y$. While this interpolation is simple and effective, the resulting score no longer corresponds to any valid probabilistic model.

Plugging the guided score equation 5 into the probability flow dynamics yields the *CFG probability flow ODE*:

$$\frac{\mathrm{d}\boldsymbol{x}_t}{\mathrm{d}t} = -\alpha(t)\boldsymbol{x}_t - \beta(t)\, \hat{\boldsymbol{s}}_t(\boldsymbol{x}_t; y, \omega). \tag{6}$$

## 3 STAGE-WISE BEHAVIOR OF CLASSIFIER FREE GUIDANCE IN SAMPLING

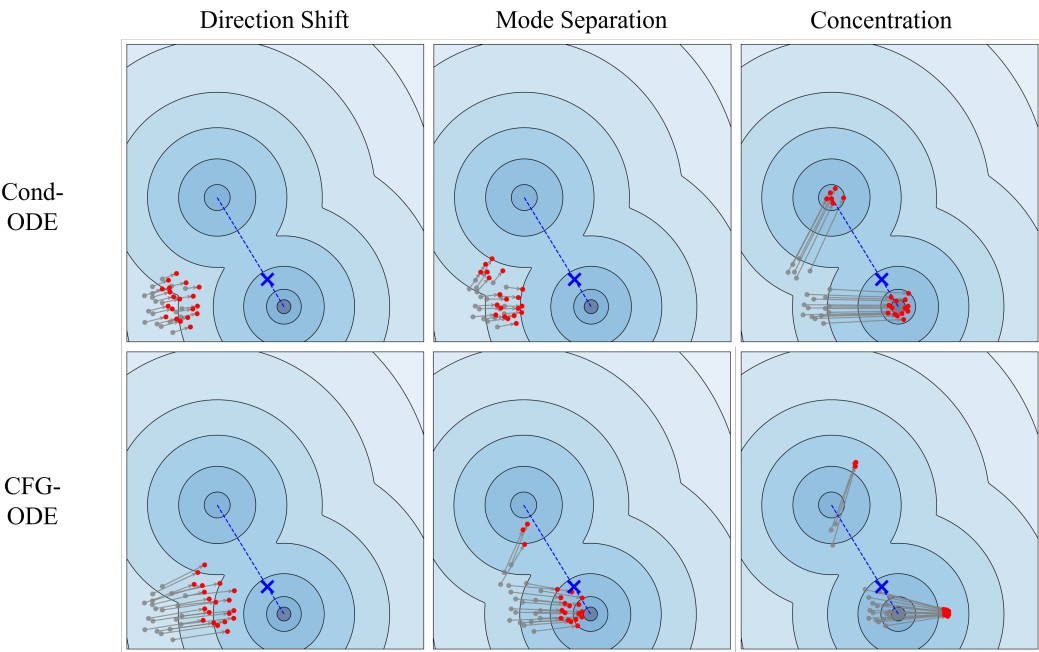

Figure 1: Illustration of the three-stage dynamics of conditional sampling (Cond-ODE, top row) versus Classifier-Free Guidance (CFG-ODE, bottom row) under a multimodal distribution. In the *Direction Shift* stage (left), CFG trajectories deviate more strongly toward the global weighted mean, introducing initialization bias. In the *Mode Separation* stage (middle), Cond-ODE trajectories maintain coverage of multiple modes, while CFG trajectories suppress weaker modes and collapse toward dominant ones. In the *Concentration* stage (right), CFG trajectories contract excessively within modes, leading to loss of fine-grained diversity. Red dots denote samples, gray arrows connect the start and end points of the same trajectory (indicating their correspondence), and blue crosses mark the weighted mean of conditional modes.

In this section, we systematically investigate the effect of Classifier-Free Guidance on sampling when the conditional distribution is multi-modal. We identify a three-stage progression of the dy-

namics: in the *Direction Shift* stage (early, high-noise regime), CFG accelerates convergence toward the class-weighted mean and induces early norm inflation; in the *Mode Separation* stage (intermediate regime), CFG primarily *accelerates* the pre-existing mode-attraction dynamics while remaining *relatively neutral* with respect to the geometry of attraction—nontrivial regions that flow to weaker modes persist and are *independent of the guidance scale*. Nevertheless, the initialization bias inherited from the first stage causes far fewer trajectories to enter these regions, so that the interaction of the first two stages leads to an effective loss of diversity. Finally, in the *Concentration* stage (late regime), CFG amplifies the restoring force toward mode centers, causing trajectories within the same mode to contract more tightly. As a result, intra-class variability is suppressed, leading to a loss of fine-grained diversity despite the samples appearing sharper and more aligned.

We begin by stating the distributional and noise-schedule assumptions that ground our analysis. These provide a simplified yet representative setting for characterizing the three stages and their interactions.

**Assumption 3.1.** Let $x \in \mathbb{R}^d$ be the data variable and $y \in \mathcal{Y}$ the condition.

**(A1) Unconditional Distribution.** $p_0(x) = \mathcal{N}(0, \mathbf{I}_d)$. This aligns with the standard Gaussian prior used in latent diffusion models, making it both realistic and analytically convenient.

**(A2) Conditional Distribution.** For each $y$, we model the conditional distribution as a Gaussian mixture:

$$p(x \mid y) = \sum_{k=1}^{K(y)} \pi_k^{(y)} \mathcal{N}(x; \boldsymbol{\mu}_k^{(y)}, \sigma_y^2 \mathbf{I}_d),$$

where $\pi_k^{(y)} \geq 0$, $\sum_k \pi_k^{(y)} = 1$, and $\sigma_y < 1$. This multimodal formulation makes it theoretically possible to analyze the impact of CFG sampling on diversity. For notational simplicity, we omit subscripts and superscripts associated with $y$ when no confusion arises.

**(A3) Noise Schedule.** In the forward SDE equation 1, we set

$$\alpha(t) = \frac{1}{1-t}, \quad \beta(t) = \frac{t}{1-t}, \quad t \in [0, 1),$$

giving

$$p(x_t \mid x_0) = \mathcal{N}\big((1-t)x_0, \, t^2 \mathbf{I}_d\big).$$

This choice corresponds to a flow-matching model with linear mean decay. We emphasize that this assumption is made for technical convenience, as different diffusion formulations can be transformed into one another through appropriate reparameterizations (Karras et al., 2022).

## 3.1 FIRST STAGE: ACCELERATION AND DIRECTION SHIFT

In the early stage (high-noise), fine-grained multimodal information is largely suppressed, so the score function reflects only global statistics of the conditional distribution rather than detailed mode structure. In this stage, Classifier-Free Guidance strengthens the global attraction and thereby alters both the *speed* and the *direction* of early trajectories by amplifying the conditional score. In practice, CFG accelerates motion toward the class-weighted mean of the mixture, while also pushing trajectories outward and inflating their norms. We make this precise in the following theorem.

**Theorem 3.2** (In-expectation early-stage proximity under CFG). *Let* $\bar{\boldsymbol{\mu}} = \sum_{k=1}^{K} \pi_k \boldsymbol{\mu}_k$ *denote the class-weighted mean of the Gaussian mixture prior, and assume the same initialization* $x_1 \sim \mathcal{N}(0, \mathbf{I})$ *for both trajectories. Then for any* $\omega > 1$*, there exists a time point* $t_{e1} < 1$ *such that, for all* $t \in [t_{e1}, 1)$*,*

$$\mathbb{E}\Big[\big\|x_t^{(\mathrm{CFG})} - \omega\bar{\boldsymbol{\mu}}\big\|_2^2\Big] < \mathbb{E}\Big[\big\|x_t^{(y)} - \omega\bar{\boldsymbol{\mu}}\big\|_2^2\Big],$$

*where* $\omega$ *is the guidance weight,* $x_t^{(y)}$ *is the solution to the conditional probability flow ODE and* $x_t^{(\mathrm{CFG})}$ *the solution to the CFG-driven ODE, both starting from* $x_1$*. In expectation, the CFG-driven trajectory remains closer to the reference point* $\omega\bar{\boldsymbol{\mu}}$ *than the purely conditional one in the early stage of sampling.*

**Interpretation.** Theorem 3.2 reveals two characteristic consequences of CFG in the high-noise regime.

First, because strong noise suppresses fine-grained structure, the score is dominated by global averages. When CFG amplifies this score, it effectively increases the pull toward the scaled mean $\omega\bar{\boldsymbol{\mu}}$, leading to an *acceleration effect*: trajectories approach the global mean faster than they would under the conditional flow alone.

Second, since $\omega > 1$ enlarges the mean by a factor, the target point $\omega\bar{\boldsymbol{\mu}}$ lies farther from the origin. Trajectories that are pulled toward this more distant point inevitably attain larger Euclidean norms, producing what we term *norm inflation*.

Together, these two effects introduce a structural bias at the very beginning of sampling. By steering trajectories rapidly toward a magnified global mean, CFG suppresses the natural exploration of multimodal variability and predisposes samples to collapse into the dominant mode once the multimodal structure becomes relevant. Thus, the first stage not only governs the pace and scale of early dynamics but also seeds the mode-selection mechanism that shapes the second stage.

## 3.2 SECOND STAGE: INTRA-CLASS MODE SEPARATION

As noise decays and the multimodal structure becomes dominant, trajectories cease to move toward the global mean and instead diverge into distinct attraction basins. This marks the onset of *mode separation*: the state space is partitioned, and each basin deterministically leads to a different mode.

In this regime, CFG plays a neutral role. Amplifying the conditional score merely *accelerates* convergence within whichever basin a trajectory already occupies, but does not reshape the basin geometry or redirect trajectories across basins. Hence, once a trajectory falls into the attraction region of a particular mode, its final outcome is determined regardless of the guidance weight.

Theorem 3.3 formalizes this neutrality in the two-component case. It establishes that the weaker mode retains a genuine basin of attraction $U_{s_2}$ that is invariant to $\omega$: any trajectory entering this region will remain aligned with the weaker mode throughout its evolution.

**Theorem 3.3** (Persistence of the weaker mode under CFG). *Consider a two-component Gaussian mixture*

$$p_0(\boldsymbol{x} \mid y) = \pi_1 \mathcal{N}(\boldsymbol{x}; \boldsymbol{\mu}_1, \sigma^2 \mathbf{I}) + \pi_2 \mathcal{N}(\boldsymbol{x}; \boldsymbol{\mu}_2, \sigma^2 \mathbf{I}), \quad \|\boldsymbol{\mu}_1\| = \|\boldsymbol{\mu}_2\|, \ \pi_1 < \pi_2.$$

*Under some mild assumptions, there exist $t_{s_2} \in (0,1)$ and an $\omega$-independent region $U_{s_2} \subset \mathbb{R}^d$, depending only on $(\pi_k, \boldsymbol{\mu}_k, \sigma)$, such that for any $\omega \geq 1$, if $\boldsymbol{x}_{t_{s_2}} \in U_{s_2}$ and $\boldsymbol{x}_t$ follows the CFG probability flow ODE equation 6 on $[0, t_{s_2}]$, then*

$$\pi_1 \mathcal{N}\big(\boldsymbol{x}_t; (1-t)\boldsymbol{\mu}_1, \ (t^2 + (1-t)^2 \sigma^2)\mathbf{I}\big) > \pi_2 \mathcal{N}\big(\boldsymbol{x}_t; (1-t)\boldsymbol{\mu}_2, \ (t^2 + (1-t)^2 \sigma^2)\mathbf{I}\big) \quad \forall\, t \in [0, t_{s_2}].$$

**Interpretation.** Theorem 3.3 shows that the weaker mode $\boldsymbol{\mu}_1$ possesses a bona fide basin of attraction $U_{s_2}$ that is independent of $\omega$. This result highlights the *neutrality* of CFG in the mode-separation regime: amplifying the conditional score does not distort the geometry of attraction basins. Once a trajectory falls into $U_{s_2}$, it will remain aligned with $\boldsymbol{\mu}_1$ throughout its evolution, no matter how large the guidance weight is chosen. Thus, the disappearance of weaker modes observed in practice cannot be ascribed to the intrinsic geometry of second-stage dynamics alone.

**Interaction with the first stage.** The missing element is the *initialization bias* accumulated during the high-noise regime. While CFG is neutral once modes separate, it rarely supplies trajectories that actually enter $U_{s_2}$. In the early stage, the amplified score consistently drags samples toward the scaled mean $\omega\bar{\boldsymbol{\mu}}$. By the time noise decays enough for the multimodal structure to emerge, most trajectories have already been displaced into regions where the drift field unequivocally favors the dominant mode $\boldsymbol{\mu}_2$. In other words, the second stage does not *destroy* weaker modes, but the first stage strongly reduces the likelihood that any trajectory remains close enough to them for mode separation to take effect. This mechanism is formalized in Proposition 3.4.

**Proposition 3.4** (Initialization bias from the first stage). *Let $\boldsymbol{x}_t$ evolve under the CFG probability flow ODE equation 6 with $\omega \geq 1$. Then there exists $0 < t_{s_1} < 1$ such that, for any $k > 1$, one can find a radius $r(k) > 0$ satisfying: if*

$$\|\boldsymbol{x}_{t_{s_1}} - k\bar{\boldsymbol{\mu}}\| < r(k),$$

*then for all $t \leq t_{s_1}$ it holds that*

$$\pi_1 \, \mathcal{N}\big(\boldsymbol{x}_t; (1-t)\boldsymbol{\mu}_1, \, (t^2 + (1-t)^2\sigma^2)\mathbf{I}\big) < \pi_2 \, \mathcal{N}\big(\boldsymbol{x}_t; (1-t)\boldsymbol{\mu}_2, \, (t^2 + (1-t)^2\sigma^2)\mathbf{I}\big).$$

*Moreover, $r(k)$ grows monotonically with $k$.*

**Interpretation.** Proposition 3.4 makes explicit how the first stage biases the sampling distribution. Once a trajectory is drawn sufficiently close to the scaled mean $k\bar{\boldsymbol{\mu}}$, the surrounding drift field guarantees that the posterior likelihood of $\boldsymbol{\mu}_2$ will dominate that of $\boldsymbol{\mu}_1$ at all earlier times. According to Theorem 3.2, a larger $\omega$ causes early-stage trajectories to be closer to $\omega\bar{\boldsymbol{\mu}}$, corresponding to a larger effective $k$ and thus expanding the region dominated by the stronger mode. Consequently, although the basin of $\boldsymbol{\mu}_1$ mathematically persists, it becomes increasingly difficult for trajectories to reach it under strong guidance. The scarcity of weaker-mode samples in practice therefore reflects not the elimination of $U_{s_2}$, but the fact that most trajectories are already preconditioned by the first stage to fall outside it.

**Stage summary.** The second stage reveals a subtle but important contrast. On a theoretical level, the dynamics preserve all modes, including weaker components, whose attraction basins remain stable and $\omega$-independent. On a practical level, however, the initialization bias carried over from the first stage severely limits access to these basins. Thus, the loss of weaker-mode diversity is not due to second-stage suppression, but to the compounded effect of early-stage displacement and reduced basin occupancy. This interplay explains why weaker modes remain viable in theory yet vanish in empirical samples.

### 3.3 THIRD STAGE: CONCENTRATION

In the final stage, when the noise level has decayed to a certain level, the fine-grained structure of the conditional distribution takes full control of the dynamics. Unlike the first stage, where global statistics dominate, or the second stage, where trajectories are separated into different basins, the third stage is governed by local geometry around each mode $\boldsymbol{\mu}_k$. In this regime, trajectories evolve almost exclusively under the influence of the nearest mode, and the role of CFG becomes entirely *within-mode*.

This yields a concentration effect: relative to standard conditional sampling ($\omega=1$), CFG-guided trajectories contract more aggressively within a basin, thereby reducing pairwise separation and diminishing within-class dispersion (fine-grained variability conditioned on a fixed semantic label). The following theorem formalizes this effect.

**Theorem 3.5** (CFG yields stronger within-mode contraction). *Under some mild assumptions, there exist a time $t_{s_3} \in (0,1)$ and a radius $r > 0$ such that the following holds uniformly. For any $k \in \{1,2\}$ and any initial pair*

$$\boldsymbol{x}_{t_{s_3}}, \, \boldsymbol{z}_{t_{s_3}} \in \mathbb{B}\big((1-t_{s_3})\,\boldsymbol{\mu}_k, \, r\big),$$

*let $\boldsymbol{x}_t^{(\mathrm{CFG})}, \boldsymbol{z}_t^{(\mathrm{CFG})}$ (resp. $\boldsymbol{x}_t^{(y)}, \boldsymbol{z}_t^{(y)}$) be the solutions initialized at the same pair $(\boldsymbol{x}_{t_{s_3}}, \boldsymbol{z}_{t_{s_3}})$ at time $t_{s_3}$ under the CFG flow with weight $\omega$ (resp. the standard conditional flow with $\omega = 1$). Then for all $t \in [0, t_{s_3})$,*

$$\big\|\boldsymbol{x}_t^{(\mathrm{CFG})} - \boldsymbol{z}_t^{(\mathrm{CFG})}\big\| < \big\|\boldsymbol{x}_t^{(y)} - \boldsymbol{z}_t^{(y)}\big\|.$$

*Here $\mathbb{B}(c,r) := \{x \in \mathbb{R}^d : \|x - c\| < r\}$ denotes the open Euclidean ball.*

**Interpretation.** Theorem 3.5 demonstrates that, under continuous dynamics, CFG strengthens the contraction of trajectories inside each mode. The intuition is straightforward: when noise is negligible, the conditional score essentially acts as a linear restoring force that points toward the local mean. Scaling this score by $\omega > 1$ increases the strength of this force, causing pairs of nearby trajectories to converge faster than they would under standard conditional sampling. Consequently, their separation shrinks more rapidly, reducing local variability.

From a generative perspective, this contraction has a dual effect. On one hand, it explains why large guidance weights often produce samples that appear sharper, cleaner, and more faithfully aligned with the conditioning signal—because trajectories are pulled tightly into the most semantically representative regions of each mode. On the other hand, it also clarifies why intra-class diversity suffers:

by collapsing trajectories together, CFG suppresses natural variations such as pose, texture, or fine-grained stylistic features, which would otherwise emerge from looser sampling around the same mode.

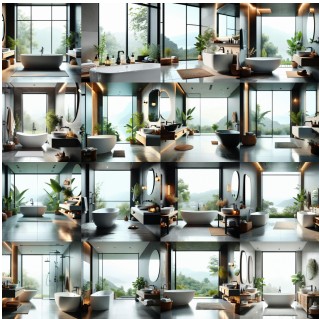 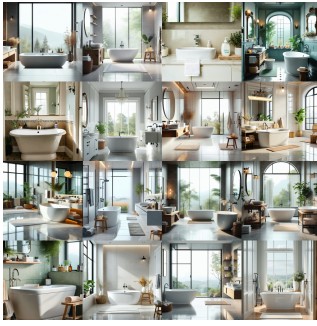 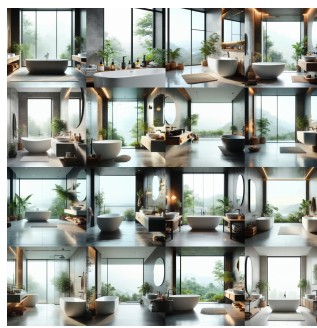

(a) Constant high weight          (b) Early low weight          (c) Early high weight

Figure 2: Comparison of guidance schedules on the prompt *"A view of a bathroom that is clean."* The **(a) Constant schedule** and **(c) Early-high schedule** both collapse diversity, with most samples converging to layouts dominated by large windows and uniform cool tones. The **(b) Early-low schedule** mitigates this effect, producing more varied spatial structures and color palettes.

### 3.4 SUMMARY ACROSS STAGES

The analysis above reveals that the effect of Classifier-Free Guidance (CFG) cannot be understood in isolation at any single point in the sampling process. Instead, its influence unfolds sequentially across three distinct stages, each of which leaves a lasting imprint on the final distribution of samples.

**Stage I: Acceleration and Direction Shift.** Under strong noise, fine-grained multimodal information is suppressed, and the score reflects only global statistics. CFG amplifies this global signal, accelerating convergence toward the class-weighted mean and inflating trajectory norms. This early bias seeds an initialization effect: samples are drawn disproportionately toward regions aligned with the global mean.

**Stage II: Mode Separation.** As noise decreases, trajectories diverge into distinct basins associated with different modes. CFG itself does not alter the geometry of attraction—basins remain intact and weaker modes preserve nontrivial regions of influence. However, because most trajectories were preconditioned by the first stage, they rarely enter the weaker basins. The disappearance of weaker-mode samples thus arises not from second-stage geometry but from the initialization bias inherited from Stage I.

**Stage III: Concentration.** Once noise becomes negligible, dynamics are dominated by local contraction within each mode. By scaling the conditional score, CFG sharpens this contraction, suppressing intra-class variability. Samples therefore appear sharper and more semantically faithful, but diversity within each mode is eroded.

Taken together, these stages explain the dual empirical effects of CFG: it improves conditional fidelity and visual sharpness, yet simultaneously diminishes both global coverage (loss of weaker modes) and local diversity (within-mode contraction). Interestingly, this resonates with prior work (Balaji et al., 2022; Li et al., 2024) that empirically found diffusion models determine the overall shape early in the process and fine details late, which corresponds to our distinction between global diversity and local diversity. This stage-wise perspective provides a unified theoretical framework for understanding diversity loss under CFG and points to principled strategies for designing time-varying guidance schedules.

## 4 EXPERIMENTS

We now turn to empirical validation of our theoretical analysis. We begin by testing two key predictions: (1) strong early guidance erodes *global* diversity, and (2) strong late guidance reduces

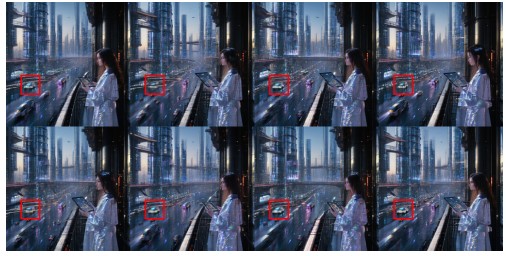 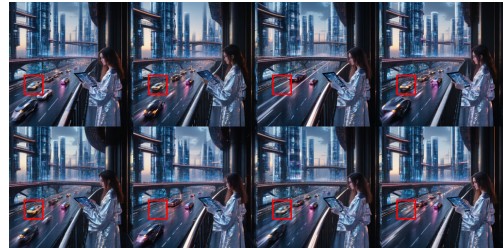

(a) Late-high schedule, MSE=0.0025      (b) Constant schedule, MSE=0.0103

Figure 3: Prompt: *Futuristic city at sunset, glass towers with neon skybridges, flying cars leaving light trails, woman in silver robe holding holographic tablet on balcony, cinematic lighting, ultra-detailed, 8K render*. Under the late-high schedule (a), the highlighted regions reveal cars that are nearly identical across samples, indicating reduced diversity. In contrast, the constant schedule (b) preserves greater variability in both car shapes and positions, which is also reflected by a larger mean squared error (MSE).

*fine-grained* diversity. We then evaluate the effectiveness of the time-varying schedule derived from theory, which weakens guidance in the early and late stages while strengthening it in the middle. Although the main contribution of this work is to explain the mechanism behind diversity loss under CFG, we also show that the proposed schedule yields tangible improvements, thereby bridging theory and practice. Detailed experimental settings are provided in the Appendix.

Table 1: Results with NFE fixed at 10 across guidance scales. Bold indicates the best value within each method at the same $\omega$ (higher is better for CLIP/IR; lower is better for FID).

| Metric | Variant | CFG | | | | APG | | | |
|---|---|---|---|---|---|---|---|---|---|
| | | $\omega=3$ | $\omega=5$ | $\omega=7$ | $\omega=9$ | $\omega=3$ | $\omega=5$ | $\omega=7$ | $\omega=9$ |
| CLIP ↑ | vanilla | **0.320** | **0.322** | **0.319** | 0.313 | **0.322** | **0.322** | **0.321** | 0.317 |
| | interval | 0.315 | 0.317 | 0.318 | **0.319** | 0.315 | 0.317 | 0.318 | **0.319** |
| | TV(Ours) | 0.319 | 0.319 | **0.319** | **0.319** | 0.319 | 0.320 | 0.320 | **0.319** |
| | $\beta$ | 0.316 | 0.318 | 0.318 | 0.316 | 0.317 | 0.318 | 0.318 | 0.317 |
| IR ↑ | vanilla | **0.894** | 0.806 | 0.553 | 0.223 | **0.909** | 0.890 | 0.712 | 0.434 |
| | interval | 0.602 | 0.694 | 0.727 | 0.723 | 0.602 | 0.693 | 0.724 | 0.725 |
| | TV(Ours) | 0.859 | **0.935** | **0.950** | **0.932** | 0.860 | **0.937** | **0.958** | **0.941** |
| | $\beta$ | 0.705 | 0.822 | 0.831 | 0.790 | 0.737 | 0.846 | 0.867 | 0.833 |
| FID ↓ | vanilla | 28.305 | 29.275 | 32.859 | 38.988 | 28.028 | 28.208 | 30.268 | 34.950 |
| | interval | 30.485 | 28.167 | **27.884** | **28.413** | 30.461 | 28.155 | **28.000** | **28.427** |
| | TV(Ours) | **27.898** | **27.722** | 28.547 | 30.259 | **27.917** | **27.935** | 28.503 | 29.838 |
| | $\beta$ | 28.679 | 28.571 | 29.774 | 32.184 | 28.163 | 28.082 | 29.148 | 31.615 |
| Saturation | vanilla | 0.283 | 0.408 | 0.513 | 0.574 | 0.266 | 0.385 | 0.500 | 0.576 |
| | interval | 0.173 | 0.178 | 0.185 | 0.193 | 0.172 | 0.177 | 0.183 | 0.192 |
| | TV(Ours) | 0.197 | 0.229 | 0.263 | 0.298 | 0.193 | 0.221 | 0.253 | 0.288 |
| | $\beta$ | 0.184 | 0.219 | 0.259 | 0.305 | 0.186 | 0.218 | 0.257 | 0.304 |
| Diversity ↑ | vanilla | 1.066 | 1.101 | 1.105 | 1.081 | 1.059 | 1.115 | 1.136 | 1.128 |
| | interval | 1.013 | 1.073 | 1.122 | 1.160 | 1.013 | 1.072 | 1.121 | 1.159 |
| | TV(Ours) | 1.092 | 1.158 | 1.196 | 1.223 | 1.088 | 1.154 | 1.194 | 1.222 |
| | $\beta$ | **1.123** | **1.181** | **1.217** | **1.242** | **1.121** | **1.178** | **1.214** | **1.241** |

## 4.1 VALIDATION OF THEORY.

We first test whether the predictions of our theoretical analysis manifest in practice. In addition to vanilla CFG with a constant high weight, we evaluate two time-varying variants: one that applies a high weight early and a low weight late, and the other with the opposite schedule. Figure 2 compares generated samples across these three strategies. Relative to vanilla CFG and the early-high variant,

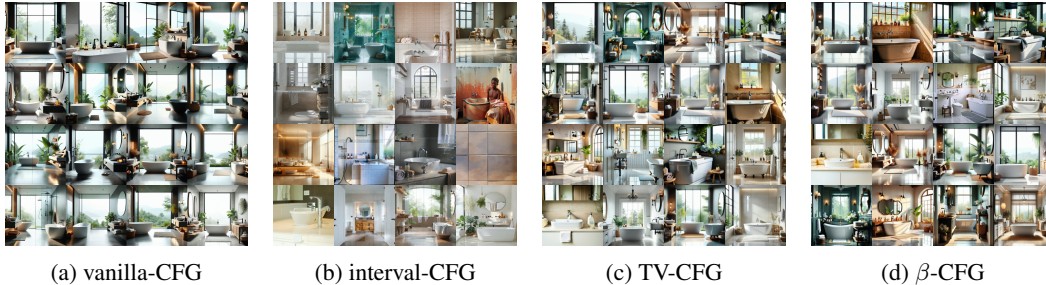

| (a) vanilla-CFG | (b) interval-CFG | (c) TV-CFG | (d) $\beta$-CFG |

Figure 4: Generated samples with the prompt *"A view of a bathroom that is clean"* at high sampling budget (NFE=50). While constant schedules yield semantically consistent but overly uniform results, methods adhering to the low-high-low scheduling principle exhibit significantly higher diversity.

the early-low variant achieves substantially higher diversity, directly supporting our first theoretical claim: excessive early guidance induces a mean-shift bias, suppressing weaker modes and thereby reducing multimodal coverage.

We next evaluate our second claim: that strong late-stage guidance enforces excessive similarity in fine details. Starting from identical noise, we integrate to an intermediate step, inject small Gaussian perturbations, and then continue sampling. One schedule keeps a low weight throughout, while the other switches to a higher weight late. As shown in Fig. 3, the late-high schedule yields outputs that are more alike in local structure, whereas the constant-low schedule preserves greater variability. This confirms that strong late guidance diminishes fine-grained diversity: once trajectories are confined to basins, large late weights amplify within-mode contraction and suppress injected perturbations, leading to nearly indistinguishable outcomes.

Table 2: Results with guidance weight fixed at $\omega = 9$ while varying the NFE budget. Bold indicates the best value within each method at the same NFE (higher is better for CLIP/IR; lower is better for FID).

| Metric | Variant | CFG | | | | APG | | | |
|---|---|---|---|---|---|---|---|---|---|
| | | NFE = 5 | NFE = 10 | NFE = 15 | NFE = 20 | NFE = 5 | NFE = 10 | NFE = 15 | NFE = 20 |
| CLIP ↑ | vanilla | 0.275 | 0.313 | **0.319** | **0.320** | 0.282 | 0.317 | **0.320** | **0.320** |
| | interval | **0.314** | **0.319** | 0.318 | 0.317 | **0.314** | **0.319** | 0.319 | 0.317 |
| | TV(Ours) | 0.312 | **0.319** | **0.319** | 0.319 | 0.312 | **0.319** | 0.319 | 0.319 |
| | $\beta$ | 0.302 | 0.316 | 0.318 | 0.318 | 0.304 | 0.317 | 0.318 | 0.318 |
| IR ↑ | vanilla | -1.137 | 0.223 | 0.616 | 0.820 | -0.926 | 0.434 | 0.735 | 0.860 |
| | interval | 0.145 | 0.723 | 0.827 | 0.863 | 0.145 | 0.725 | 0.825 | 0.865 |
| | TV(Ours) | **0.176** | **0.932** | **1.016** | **1.049** | **0.174** | **0.941** | **1.029** | **1.061** |
| | $\beta$ | -0.328 | 0.790 | 0.971 | 1.024 | -0.202 | 0.833 | 0.984 | 1.036 |
| FID ↓ | vanilla | 80.482 | 38.988 | 31.863 | 29.059 | 73.751 | 34.950 | 30.122 | 28.602 |
| | interval | 47.239 | **28.413** | **25.747** | **25.385** | 47.735 | **28.427** | **25.691** | **25.365** |
| | TV(Ours) | **46.895** | 30.259 | 29.256 | 28.458 | **46.878** | 29.838 | 28.834 | 27.965 |
| | $\beta$ | 62.979 | 32.184 | 29.246 | 28.348 | 58.398 | 31.615 | 28.836 | 27.968 |
| Saturation | vanilla | 0.589 | 0.574 | 0.526 | 0.476 | 0.610 | 0.576 | 0.521 | 0.472 |
| | interval | 0.197 | 0.193 | 0.212 | 0.233 | 0.194 | 0.192 | 0.209 | 0.229 |
| | TV(Ours) | 0.271 | 0.298 | 0.325 | 0.332 | 0.268 | 0.288 | 0.309 | 0.311 |
| | $\beta$ | 0.236 | 0.305 | 0.316 | 0.330 | 0.243 | 0.304 | 0.307 | 0.315 |
| Diversity ↑ | vanilla | 0.867 | 1.081 | 1.171 | 1.201 | 0.917 | 1.128 | 1.186 | 1.200 |
| | interval | 1.140 | 1.160 | 1.178 | 1.202 | 1.140 | 1.159 | 1.176 | 1.200 |
| | TV(Ours) | 1.191 | 1.223 | 1.233 | 1.231 | 1.191 | 1.222 | 1.232 | 1.231 |
| | $\beta$ | **1.196** | **1.242** | **1.241** | **1.239** | **1.198** | **1.241** | **1.239** | **1.238** |

## 4.2 METHODOLOGICAL IMPLICATIONS.

Guided by the above analysis, we propose a linear time-varying schedule where the guidance weight rises early, peaks at the intermediate stage, and decreases late (TV-CFG). We also considered Interval-CFG (Kynkäänniemi et al., 2024) and $\beta$-CFG methods (Malarz et al., 2025). Both adhere to the time-varying scheduling principle of being weak in the early stages, strong in the middle, and

weak in the late stages, which theoretically contributes to sampling diversity. The specific scheduling principles are detailed in the Appendix. Though not our main focus, this theory-derived design yields empirical gains and highlights how stage-wise insights translate into practical improvements in robustness and diversity.

Figure 4 shows that our schedule preserves diversity while reducing over-saturation. Table 1 further reports consistent gains in ImageReward (IR) (Xu et al., 2024), which better reflects overall quality than CLIP. Notably, all three methods that follow our design principle effectively improve generation diversity while maintaining good generation quality. At low weights our method matches vanilla CFG, but at *high weights*—where vanilla CFG degrades—it achieves clear advantages and stronger best-case performance. The same principle applies to APG (Sadat et al., 2025), where TV-APG likewise improves IR.

Table 2 summarizes results across NFEs. At low NFEs, vanilla CFG nearly collapses, producing poor IR and FID; at higher NFEs it partially recovers but still suffers from over-saturation, with high saturation values and limited diversity gains. We attribute this to norm inflation from strong early guidance under coarse discretization, which destabilizes dynamics and cannot be fully corrected by increasing NFEs. Thus, theory-guided schedules are particularly effective in low- and medium-NFE regimes, where budgets are limited and naive CFG is most fragile.

## 5 CONCLUSION

We present the first systematic analysis of Classifier-Free Guidance (CFG) under a multimodal conditional distribution assumption, characterizing its dynamics in three stages: early *Direction Shift*, mid *Mode Separation*, and late *Concentration*. Our theory explains how diversity deteriorates: early mean-shift bias suppresses weaker modes, while late-stage contraction reduces intra-class variability. This stage-wise perspective clarifies long-standing empirical observations and provides concrete predictions consistent with behaviors observed in modern diffusion models. Beyond theoretical analysis, we further show that attenuating guidance in the early and late stages while emphasizing the mid stage offers a simple yet effective strategy for mitigating diversity loss. Finally, extending the stage-wise framework to stochastic differential equation (SDE) sampling and rigorously characterizing the effects of numerical discretization remain important directions for future research.

## 6 ACKNOWLEDGEMENTS

The authors are with the Department of Electronic Engineering, Beijing National Research Center for Information Science and Technology, Tsinghua University, Beijing 100084, China. This work was supported by the National Key Research and Development Program of China (Grant No. 2025YFF0515601) and the National Natural Science Foundation of China (NSAF U2230201).

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

## A LLM USAGE

We used large language models (LLMs) solely for language polishing. All research ideas, theoretical analyses, experiment designs, and results were developed entirely by the authors. The LLMs did not contribute to ideation or scientific content.

## B RELATED WORK

Although Classifier-Free Guidance (CFG) was introduced early and has since become a standard sampling strategy for diffusion models, theoretical analyses of CFG only began to emerge around 2024. Most existing studies focus on simplified distributional settings, such as one-dimensional or Gaussian cases.

Chidambaram et al. (2024) show that in a one-dimensional setting where the unconditional distribution is bimodal and the conditional distribution is unimodal, CFG-ODE sampling tends to concentrate on the distribution's edges as the guidance weight increases. Moreover, even small score estimation errors can cause severe deviations from the target support under large guidance weights. Extending to higher dimensions, Pavasovic et al. (2025) analyze a two-component Gaussian mixture and demonstrate that such edge-concentration effects vanish in high dimensions.

Bradley & Nakkiran (2024) prove that when both conditional and unconditional distributions are one-dimensional zero-mean Gaussians, the closed-form solution of CFG deviates from the expected gamma-weighted distribution. Xia et al. (2024) further extend this result to high-dimensional isotropic Gaussians with differing conditional and unconditional parameters, deriving a closed-form output that again diverges from the gamma-weighted distribution. Li et al. (2025b) consider the effect of the covariance structure of Gaussian distributions on the CFG sampling process, and point out that CFG enhances generation quality by amplifying class-specific features while suppressing generic ones.

Building on these findings, Wu et al. (2024) show that when the unconditional distribution is a mixture of approximately orthogonal Gaussians and the conditional distribution is Gaussian, CFG sampling theoretically increases classification confidence while simultaneously decreasing the entropy of the output distribution. The entropy reduction phenomenon is consistent with our analysis of the third-stage dynamics of CFG sampling, where strong guidance induces distributional contraction and suppresses inter-modal variability. Jin et al. (2025) further relax these assumptions to more general Gaussian mixture settings, and reveal that CFG induces norm inflation and anomalous diffusion effects, thereby providing a theoretical explanation for the over-saturation phenomena observed in practice. Finally, Li & Jiao (2025) analyze CFG under considerably weaker assumptions without restricting the distributional form, and demonstrate that the improvement in classification confidence holds only in an average sense, rather than uniformly across all initial conditions.

In summary, with the exception of Li & Jiao (2025), existing works primarily focus on settings where the conditional distribution is unimodal. This restriction limits their ability to explain critical phenomena such as the loss of diversity under large guidance weights. Although Li & Jiao (2025) relax the distributional assumptions considerably, their overly general setting similarly fails to capture the underlying mechanism of the phenomenon.

## C PROOF OF THEOREM AND PROPOSITION

We begin by stating a key lemma, which serves as a fundamental tool throughout the subsequent analysis.

**Lemma C.1.** *For the probability flow ODE associated with the forward noising process, the dynamics of $\boldsymbol{x}_t$ satisfy*

$$\frac{\mathrm{d}\boldsymbol{x}_t}{\mathrm{d}t} = -\frac{\mathbb{E}[\boldsymbol{x}_0 \mid \boldsymbol{x}_t] - \boldsymbol{x}_t}{t}. \tag{7}$$

*In the unconditional setting where the prior is isotropic Gaussian, the posterior mean admits the closed-form expression*

$$\hat{\boldsymbol{x}}_{0|t} := \mathbb{E}[\boldsymbol{x}_0 \mid \boldsymbol{x}_t] = \frac{1-t}{t^2 + (1-t)^2} \boldsymbol{x}_t. \tag{8}$$

*In the conditional case where the data distribution is modeled as a Gaussian mixture, the posterior mean can be expressed as a convex combination*

$$\hat{\boldsymbol{x}}_{0|t}^{(y)} := \mathbb{E}[\boldsymbol{x}_0 \mid \boldsymbol{x}_t, y] \;=\; \sum_{k=1}^{K} \tilde{\pi}_k(\boldsymbol{x}_t)\, \boldsymbol{m}_k(\boldsymbol{x}_t), \tag{9}$$

*where the component-wise posterior mean and responsibilities are given respectively by*

$$\boldsymbol{m}_k(\boldsymbol{x}_t) = \frac{t^2\, \boldsymbol{\mu}_k + (1-t)\sigma^2 \boldsymbol{x}_t}{(1-t)^2\sigma^2 + t^2}, \tag{10}$$

$$\tilde{\pi}_k(\boldsymbol{x}_t) = \frac{\pi_k\, \mathcal{N}\Big(\boldsymbol{x}_t;\, (1-t)\boldsymbol{\mu}_k,\, \big((1-t)^2\sigma^2 + t^2\big)\mathbf{I}\Big)}{\sum_j \pi_j\, \mathcal{N}\Big(\boldsymbol{x}_t;\, (1-t)\boldsymbol{\mu}_j,\, \big((1-t)^2\sigma^2 + t^2\big)\mathbf{I}\Big)}. \tag{11}$$

The first identity follows directly from Tweedie's formula, while the latter two expressions are obtained via straightforward posterior computations for Gaussian and Gaussian mixture distributions.

## C.1 PROOF OF THEOREM 3.2

**Theorem C.2** (Theorem 3.2). *Assume the class-conditional prior is a $K$-component Gaussian mixture with shared covariance $\sigma^2\mathbf{I}$ and weights $\{\pi_k\}_{k=1}^{K}$, and let $\bar{\boldsymbol{\mu}} = \sum_k \pi_k \boldsymbol{\mu}_k$. Let the reverse probability flow ODE and its CFG-driven variant be driven by the estimators from Lemma 1:*

$$\dot{\boldsymbol{x}}_t^{(y)} \;=\; \frac{\hat{\boldsymbol{x}}_{0|t}^{(y)}(\boldsymbol{x}_t^{(y)}) - \boldsymbol{x}_t^{(y)}}{t}, \tag{12}$$

$$\dot{\boldsymbol{x}}_t^{(\mathrm{CFG})} \;=\; \frac{\hat{\boldsymbol{x}}_{0|t}^{(y)}(\boldsymbol{x}_t^{(\mathrm{CFG})}) + \omega\Big(\hat{\boldsymbol{x}}_{0|t}^{(y)}(\boldsymbol{x}_t^{(\mathrm{CFG})}) - \hat{\boldsymbol{x}}_{0|t}(\boldsymbol{x}_t^{(\mathrm{CFG})})\Big) - \boldsymbol{x}_t^{(\mathrm{CFG})}}{t}, \tag{13}$$

*with a shared random initialization $\boldsymbol{x}_1^{(y)} = \boldsymbol{x}_1^{((\mathrm{CFG}))} \overset{d}{=} \mathcal{N}(\mathbf{0}, \mathbf{I})$ independent of $y$. Then for any $\omega > 1$ there exists $t_{e1} \in (0,1)$ such that for all $t \in [t_{e1}, 1)$,*

$$\mathbb{E}\Big[\big\|\boldsymbol{x}_t^{(\mathrm{CFG})} - \omega\bar{\boldsymbol{\mu}}\big\|_2^2 - \big\|\boldsymbol{x}_t^{(y)} - \omega\bar{\boldsymbol{\mu}}\big\|_2^2\Big] \;<\; 0.$$

**Proof Sketch.** By Lemma C.1, as $t \to 1^-$ we have the uniform early-time approximations $\hat{\boldsymbol{x}}_{0|t}^{(y)}(\boldsymbol{x}) \approx \bar{\boldsymbol{\mu}}$ and $\hat{\boldsymbol{x}}_{0|t}(\boldsymbol{x}) \approx 0$. Hence the two drifts point toward $\bar{\boldsymbol{\mu}}$ (no-CFG) and $\omega\bar{\boldsymbol{\mu}}$ (CFG), respectively. Using the Lyapunov functional $V(\boldsymbol{x}) = \|\boldsymbol{x} - \omega\bar{\boldsymbol{\mu}}\|^2$, the initial gap derivative satisfies $D'(1^-) > 0$, so by continuity there exists $t_{e1} \in (0,1)$ with $D(t) < 0$ for all $t \in [t_{e1}, 1)$. Thus, in the early regime, the CFG trajectory is (in expectation) closer to $\omega\bar{\boldsymbol{\mu}}$ than the unguided one.

*Proof.* Define the quadratic functional $V(\boldsymbol{x}) = \|\boldsymbol{x} - \omega\bar{\boldsymbol{\mu}}\|_2^2$ and the gap

$$D(t) \;:=\; \mathbb{E}\Big[V\big(\boldsymbol{x}_t^{(\mathrm{CFG})}\big) - V\big(\boldsymbol{x}_t^{(y)}\big)\Big].$$

Clearly $D(1) = 0$ since the two processes share the same random initial condition $\boldsymbol{x}_1$. We first compute the time derivative of $D$. Under the standard local-Lipschitz and linear-growth conditions satisfied by the fields in Lemma 1 (see below), both trajectories have uniformly bounded second moments on $t \in [t_0, 1]$ and differentiation under the expectation is justified by dominated convergence:

$$D'(t) \;=\; \mathbb{E}\Big[\frac{\mathrm{d}}{\mathrm{d}t}V(\boldsymbol{x}_t^{(\mathrm{CFG})})\Big] - \mathbb{E}\Big[\frac{\mathrm{d}}{\mathrm{d}t}V(\boldsymbol{x}_t^{(y)})\Big] \;=\; \mathbb{E}[\mathcal{L}_{\mathrm{cfg}}V\big(t, \boldsymbol{x}_t^{(\mathrm{CFG})}\big)] \;-\; \mathbb{E}[\mathcal{L}_y V\big(t, \boldsymbol{x}_t^{(y)}\big)],$$

where $\nabla V(\boldsymbol{x}) = 2(\boldsymbol{x} - \omega\bar{\boldsymbol{\mu}})$ and

$$\mathcal{L}_y V(t, \boldsymbol{x}) = \left\langle 2(\boldsymbol{x} - \omega\bar{\boldsymbol{\mu}}), -\frac{\hat{\boldsymbol{x}}_{0|t}^{(y)}(\boldsymbol{x}) - \boldsymbol{x}}{t} \right\rangle,$$

$$\mathcal{L}_{\mathrm{cfg}} V(t, \boldsymbol{x}) = \left\langle 2(\boldsymbol{x} - \omega\bar{\boldsymbol{\mu}}), -\frac{\hat{\boldsymbol{x}}_{0|t}^{(y)}(\boldsymbol{x}) + \omega(\hat{\boldsymbol{x}}_{0|t}^{(y)}(\boldsymbol{x}) - \hat{\boldsymbol{x}}_{0|t}(\boldsymbol{x})) - \boldsymbol{x}}{t} \right\rangle.$$

**Early-time expansions.** By the closed forms in Lemma 1, for any bounded set there exist $c > 0$ and $t_\circ < 1$ such that uniformly for $\|x\|$ bounded and $t \in [t_\circ, 1)$,

$$\hat{x}_{0|t}^{(y)}(x) = \bar{\mu} + r_y(t, x), \qquad \hat{x}_{0|t}(x) = r_0(t, x), \qquad \|r_y(t, x)\| + \|r_0(t, x)\| \leq c(1-t)(1+\|x\|).$$
(14)

Hence for such $(t, x)$,

$$\dot{x}_t^{(y)} = \frac{\bar{\mu} - x}{t} + O(1-t), \qquad \dot{x}_t^{(\text{CFG})} = \frac{(1+\omega)\bar{\mu} - x}{t} + O(1-t),$$
(15)

where the $O(1-t)$ terms are uniform on bounded sets as $t \to 1$.

**The limiting derivative at $t = 1$.** Using equation 15 and the fact that $x_t^{(\cdot)} \to x_1$ in $L^2$ as $t \to 1$, we obtain

$$\lim_{t \to 1^-} D'(t) = \mathbb{E}[\langle 2(x_1 - \omega\bar{\mu}), -((1+\omega)\bar{\mu} - x_1)\rangle] - \mathbb{E}[\langle 2(x_1 - \omega\bar{\mu}), -(\bar{\mu} - x_1)\rangle].$$

Expanding the inner products and using $\mathbb{E}[x_1] = 0$, we get

$$\lim_{t \to 1^-} D'(t) = -2\omega\,\mathbb{E}[\langle x_1, \bar{\mu}\rangle] + 2\omega^2\|\bar{\mu}\|_2^2 = 2\omega^2\|\bar{\mu}\|_2^2 > 0.$$

**Strict negativity on a full interval.** By continuity of all terms in equation 14–equation 15 and uniform integrability (bounded second moments), there exists $t_{e1} \in (0, 1)$ and a constant $c_\star > 0$ such that

$$D'(t) \geq c_\star > 0, \qquad \forall t \in [t_{e1}, 1).$$

Since $D(1) = 0$ and $D'$ is strictly positive on $[t_{e1}, 1)$, we integrate backward from 1 to conclude that

$$D(t) = -\int_t^1 D'(s)\,\mathrm{d}s < 0, \qquad \forall t \in [t_{e1}, 1).$$

This is exactly the desired inequality in expectation:

$$\mathbb{E}\left[\|x_t^{(\text{CFG})} - \omega\bar{\mu}\|_2^2 - \|x_t^{(y)} - \omega\bar{\mu}\|_2^2\right] = D(t) < 0.$$

$\square$

*Remark* C.3. The proof only uses that $\hat{x}_{0|t}^{(y)}(x) \to \bar{\mu}$ and $\hat{x}_{0|t}(x) \to 0$ as $t \to 1$ uniformly on compact sets, together with bounded second moments of $x_t$. Hence the statement extends to other conditional models admitting the same early-time limits.

## C.2 PROOF OF THEOREM 3.3

**Theorem C.4** (Theorem 3.3). *Consider a two-component Gaussian mixture*

$$p_0(x \mid y) = \pi_1\,\mathcal{N}(x; \mu_1, \sigma^2\mathbf{I}) + \pi_2\,\mathcal{N}(x; \mu_2, \sigma^2\mathbf{I}), \quad \|\mu_1\| = \|\mu_2\|, \ \pi_1 < \pi_2.$$

*Under mild assumptions, there exist $t_{s_2} \in (0, 1)$ and an $\omega$-independent region $U_{s_2} \subset \mathbb{R}^d$, depending only on $(\pi_k, \mu_k, \sigma)$, such that for any $\omega \geq 1$, if $x_{t_{s_2}} \in U_{s_2}$ and $x_t$ follows the CFG probability flow ODE equation 6 on $[0, t_{s_2}]$, then*

$$\pi_1\,\mathcal{N}(x_t; (1-t)\mu_1, (t^2 + (1-t)^2\sigma^2)\mathbf{I}) > \pi_2\,\mathcal{N}(x_t; (1-t)\mu_2, (t^2 + (1-t)^2\sigma^2)\mathbf{I}) \quad \forall t \in [0, t_{s_2}].$$

**Proof Sketch.** To eliminate the effect of scaling, we map the sampling trajectory to

$$\frac{x_t}{1-t},$$

which corresponds to the VP-SDE parameterization of the diffusion model. Consider a separating hyperplane

$$S_t := \{x \in \mathbb{R}^d : \langle x - c_t, \mu_1 - \mu_2\rangle = 0\},$$

lying between $\boldsymbol{\mu}_1$ and $\boldsymbol{\mu}_2$, orthogonal to $\Delta\boldsymbol{\mu} := \boldsymbol{\mu}_1 - \boldsymbol{\mu}_2$, and positioned closer to the weaker class $\boldsymbol{\mu}_1$. By construction, the conditional ODE vector field on $S_t$ is orthogonal to $\Delta\boldsymbol{\mu}$.

This hyperplane $S_t$ has three key properties: 1. On $S_t$, the CFG-ODE drift points toward the weaker class $\boldsymbol{\mu}_1$. 2. As $t$ decreases, the location $\boldsymbol{c}_t$ of $S_t$ moves toward the stronger class $\boldsymbol{\mu}_2$. 3. The two regions divided by $S_t$ differ in posterior mass: the side containing $\boldsymbol{\mu}_1$ always assigns higher responsibility to the weaker component.

Combining these properties, we conclude that if a point $\boldsymbol{x}_{t_0}$ lies in the region of $S_t$ containing $\boldsymbol{\mu}_1$ at some time $t_0$, then the trajectory $\boldsymbol{x}_t$ will remain in that region for all $t \leq t_0$.

*Proof.* Let $\Delta\boldsymbol{\mu} := \boldsymbol{\mu}_1 - \boldsymbol{\mu}_2$ and $\sigma_t^2 := t^2 + (1-t)^2\sigma^2$. Introduce the rescaled state $\boldsymbol{z}_t := \boldsymbol{x}_t/(1-t)$ and the scalar coordinate

$$u(\boldsymbol{z}) := \langle \boldsymbol{z}, \Delta\boldsymbol{\mu} \rangle.$$

Then we prove that for

**Dynamics along $\Delta\boldsymbol{\mu}$.** Under the CFG probability–flow ODE,

$$\dot{\boldsymbol{x}}_t = -\frac{1}{t}\big(\hat{\boldsymbol{x}}_{0|t}^{(\mathrm{CFG})}(\boldsymbol{x}_t) - \boldsymbol{x}_t\big), \qquad \hat{\boldsymbol{x}}_{0|t}^{(y)} = (1-\omega)\hat{\boldsymbol{x}}_{0|t} + \omega\,\hat{\boldsymbol{x}}_{0|t}^{(y)}.$$

For isotropic Gaussians, Tweedie's identity yields the shared affine form

$$\hat{\boldsymbol{x}}_{0|t}^{(y)}(\boldsymbol{x}) = a_t\,\bar{\boldsymbol{\mu}}_t(\boldsymbol{x}) + b_t\,\boldsymbol{x}, \qquad a_t = \frac{t^2}{\sigma_t^2}, \quad b_t = \frac{\sigma^2(1-t)}{\sigma_t^2},$$

where $\bar{\boldsymbol{\mu}}_t(\boldsymbol{x}) = r_1(t,\boldsymbol{x})\boldsymbol{\mu}_1 + r_2(t,\boldsymbol{x})\boldsymbol{\mu}_2$ and $r_k(t,\boldsymbol{x}) \propto \pi_k \mathcal{N}(\boldsymbol{x}; (1-t)\boldsymbol{\mu}_k, \sigma_t^2\mathbf{I})$, $r_1 + r_2 = 1$. Setting $\boldsymbol{x}_t = (1-t)\boldsymbol{z}_t$ and simplifying gives

$$\dot{\boldsymbol{z}}_t^{(y)} = \frac{a_t}{t(1-t)}\big(\boldsymbol{z}_t - \bar{\boldsymbol{\mu}}_t(\boldsymbol{x}_t)\big).$$

$$\dot{\boldsymbol{z}}_t = \frac{t\boldsymbol{z}_t}{(1-t)\big(t^2 + (1-t)^2\big)}.$$

Taking the inner product with $\Delta\boldsymbol{\mu}$ and using the identity $\frac{\boldsymbol{\mu}_1+\boldsymbol{\mu}_2}{2} \cdot \Delta\boldsymbol{\mu} = 0$, we define

$$g_t(u) := u - \frac{\|\Delta\boldsymbol{\mu}\|^2}{2}\,\tanh\!\left(\frac{(1-t)^2}{2\sigma_t^2}\big(u - c(t)\big)\right),$$

so that

$$\langle \dot{\boldsymbol{z}}_t^{(y)}, \Delta\boldsymbol{\mu} \rangle = g_t\big(u(\boldsymbol{z}_t)\big), \tag{16}$$

where

$$c(t) := \psi(t)\log\frac{\pi_2}{\pi_1}, \qquad \psi(t) := \frac{\sigma_t^2}{(1-t)^2}.$$

**Zero-thrust hyperplane and $U_t$.** The zero-thrust hyperplane at time $t$ is

$$S_t := \{\boldsymbol{z} : g_t(u(\boldsymbol{z})) = 0, u(\boldsymbol{z}) > 0, \langle \boldsymbol{z} - \boldsymbol{\mu}_1, \boldsymbol{z} - \boldsymbol{\mu}_2 \rangle < 0\}.$$

or equivalently

$$S_t = \{\boldsymbol{z} : u(\boldsymbol{z}) = e_t\}, \quad e_t > 0,$$

for some $e_t$. In other words, $S_t$ lies between $\boldsymbol{\mu}_2$ and $\boldsymbol{\mu}_1$, positioned closer to $\boldsymbol{\mu}_1$, and it is precisely on this hyperplane that the conditional velocity field has zero projection onto the direction $\Delta\boldsymbol{\mu}$.

$$U_t := \Big\{\boldsymbol{z} : u(\boldsymbol{z}) > e_t\Big\},$$

i.e., the side of $S_t$ that contains $\boldsymbol{\mu}_1$.

**Claim A (direction of the CFG field on $S_t$).** On $S_t$ we have

$$\left\langle \dot{z}_t^{(\text{CFG})}, \Delta\boldsymbol{\mu} \right\rangle\Big|_{\dot{z}_t^{(\text{CFG})} \in S_t} = \frac{\omega a_t}{t(1-t)}\, g_t(u(\boldsymbol{z}_t^{(\text{CFG})})) - (\omega-1)\frac{t}{(1-t)\big(t^2+(1-t)^2\big)} u(\boldsymbol{z}_t^{(\text{CFG})})$$

$$= 0 - (\omega-1)\frac{t}{(1-t)\big(t^2+(1-t)^2\big)} u(\boldsymbol{z}_t^{(\text{CFG})}) < 0. \tag{17}$$

where $u(\boldsymbol{z}_t^{(\text{CFG})}) > 0$ is because $\langle \boldsymbol{z}, \Delta\boldsymbol{\mu} \rangle > 0$ for $\boldsymbol{z} \in S_t$.

In words, on $S_t$ the CFG vector field points strictly toward the weaker mode $\boldsymbol{\mu}_1$ (in backward time).

**Claim B (motion of $S_t$).** We define

$$S_t := \Big\{\boldsymbol{z}: \ g_t(u(\boldsymbol{z})) = 0, \ \langle \boldsymbol{z}, \Delta\boldsymbol{\mu} \rangle > 0, \ \langle \boldsymbol{z} - \boldsymbol{\mu}_1, \boldsymbol{z} - \boldsymbol{\mu}_2 \rangle < 0 \Big\},$$

or equivalently

$$S_t = \{\boldsymbol{z}: \ u(\boldsymbol{z}) = e_t\}, \quad e_t > 0.$$

It follows that $e_t$ satisfies

$$e_t - \frac{\|\Delta\boldsymbol{\mu}\|^2}{2}\, \tanh\!\left(\frac{1}{2\big(\sigma^2 + (\frac{t}{1-t})^2\big)}\, e_t - \log\frac{\pi_2}{\pi_1}\right) = 0.$$

Since $\sigma^2 + (\frac{t}{1-t})^2$ is strictly increasing in $t$, we conclude that

$$\frac{\mathrm{d}e_t}{\mathrm{d}t} > 0.$$

In other words, as $t$ decreases (i.e., in backward time), the hyperplane $S_t$ moves closer to the stronger mode $\boldsymbol{\mu}_2$.

**Backward invariance and conclusion.** Thus, by the moving–set viability (Nagumo) criterion applied to the time-reversed system, $\{U_t\}_{t \in [0, t_{s_2}]}$ is backward invariant: if $\boldsymbol{z}_{t_{s_2}} \in U_{t_{s_2}}$, then $\boldsymbol{z}_t \in U_t$ for all $t \in [0, t_{s_2}]$. Finally, $\boldsymbol{z}_t \in U_t$ implies $u_t > c(t)$, hence

$$\pi_1\, \mathcal{N}\big(\boldsymbol{x}_t; (1-t)\boldsymbol{\mu}_1, \sigma_t^2\mathbf{I}\big) > \pi_2\, \mathcal{N}\big(\boldsymbol{x}_t; (1-t)\boldsymbol{\mu}_2, \sigma_t^2\mathbf{I}\big),$$

which proves the theorem.

*Remark* C.5. The mild condition mentioned in the theorem refers to the existence of $S_t$. In fact, for sufficiently small $t$ and $\omega$, such a set always exists. To see this, consider the regime where $t \to 0$ and $\sigma \to 0$, while the distance between $\boldsymbol{\mu}_1$ and $\boldsymbol{\mu}_2$ remains non-negligible but not too large. In this case we have the approximation

$$g_t(u) \approx u - \frac{\|\Delta\boldsymbol{\mu}\|^2}{2}, \quad \text{for all } u > 0.$$

Evaluating at $u = u(\boldsymbol{\mu}_1)$ gives

$$g_t\big(u(\boldsymbol{\mu}_1)\big) \approx \langle \boldsymbol{\mu}_1, \Delta\boldsymbol{\mu} \rangle - \frac{\|\Delta\boldsymbol{\mu}\|^2}{2} > 0,$$

while for sufficiently small $u$ we have $g_t(u) < 0$. Hence, by continuity, there exists some $e_t$ such that

$$u(\boldsymbol{z}_t) = e_t, \quad 0 < e_t < \langle \boldsymbol{\mu}_1, \boldsymbol{\mu} \rangle.$$

This ensures the non-emptiness of $S_t$ in the considered regime.

$\square$

## C.3 PROOF OF PROPOSITION 3.4

**Proposition C.6** (Proposition 3.4). *Let $\boldsymbol{x}_t$ evolve according to the CFG probability flow ODE equation 6 with guidance weight $\omega \geq 1$. Then there exists a time $0 < t_{s_1} < 1$ such that, for any $k > 1$, one can find a radius $r(k) > 0$ with the following property: if*

$$\|\boldsymbol{x}_{t_{s_1}} - k\bar{\boldsymbol{\mu}}\| < r(k),$$

*then for all $t \leq t_{s_1}$ it holds that*

$$\pi_1\, \mathcal{N}\big(\boldsymbol{x}_t; (1-t)\boldsymbol{\mu}_1, \, (t^2 + (1-t)^2\sigma^2)\mathbf{I}\big) < \pi_2\, \mathcal{N}\big(\boldsymbol{x}_t; (1-t)\boldsymbol{\mu}_2, \, (t^2 + (1-t)^2\sigma^2)\mathbf{I}\big).$$

*Moreover, the radius $r(k)$ grows monotonically with $k$.*

**Proof Sketch.** Consider the hyperplane

$$H := \{\boldsymbol{x} \in \mathbb{R}^d : \langle \boldsymbol{x}, \, \boldsymbol{\mu}_1 - \boldsymbol{\mu}_2 \rangle = 0\},$$

which is orthogonal to the vector $\Delta\boldsymbol{\mu} := \boldsymbol{\mu}_1 - \boldsymbol{\mu}_2$. One can show that along $H$, the probability–flow vector field always points toward the stronger mode $\boldsymbol{\mu}_2$. Moreover, $H$ partitions the space into two half-spaces, and the one containing $\boldsymbol{\mu}_2$ consistently assigns larger posterior weight to the stronger component. Therefore, any trajectory initialized near $k\boldsymbol{\mu}_2$ that remains on the $\boldsymbol{\mu}_2$ side of $H$ can never cross into the weaker side. Equivalently, as long as a point does not exceed this separating boundary, it will remain in the region dominated by $\boldsymbol{\mu}_2$ for all future times.

*Proof.* Let $\Delta\boldsymbol{\mu} := \boldsymbol{\mu}_2 - \boldsymbol{\mu}_1$ and $\bar{\boldsymbol{\mu}} = \pi_1 \boldsymbol{\mu}_1 + \pi_2 \boldsymbol{\mu}_2$ with $\pi_2 > \pi_1$. Fix $k > 1$ and define

$$r(k) := \frac{k \, \bar{\boldsymbol{\mu}}^\top \Delta\boldsymbol{\mu}}{\|\Delta\boldsymbol{\mu}\|}.$$

Consider any $\boldsymbol{x}$ with $\|\boldsymbol{x} - k\bar{\boldsymbol{\mu}}\| < r(k)$. Then

$$\boldsymbol{x}^\top \Delta\boldsymbol{\mu} \;\geq\; k \, \bar{\boldsymbol{\mu}}^\top \Delta\boldsymbol{\mu} - \|\boldsymbol{x} - k\bar{\boldsymbol{\mu}}\| \, \|\Delta\boldsymbol{\mu}\| \;>\; k \, \bar{\boldsymbol{\mu}}^\top \Delta\boldsymbol{\mu} - r(k)\|\Delta\boldsymbol{\mu}\| \;=\; 0.$$

Thus $B_{r(k)}(k\bar{\boldsymbol{\mu}}) \subset \{\boldsymbol{x} : \boldsymbol{x}^\top \Delta\boldsymbol{\mu} > 0\}$, i.e., the entire ball lies strictly inside the half-space where the second Gaussian component dominates.

Next, note that on the boundary $\{\boldsymbol{x} : \boldsymbol{x}^\top \Delta\boldsymbol{\mu} = 0\}$, the CFG dynamics satisfy

$$\dot{\boldsymbol{x}}_t^\top \Delta\boldsymbol{\mu} > 0,$$

which means the flow points strictly inward relative to the half-space $\{\boldsymbol{x} : \boldsymbol{x}^\top \Delta\boldsymbol{\mu} > 0\}$. By Nagumo's condition, this half-space is positively invariant under the dynamics. Therefore, if $\boldsymbol{x}_{t_{s_1}} \in B_{r(k)}(k\bar{\boldsymbol{\mu}})$ for some $t_{s_1}$ close to 1, then $\boldsymbol{x}_t$ remains in the region $\{\boldsymbol{x} : \boldsymbol{x}^\top \Delta\boldsymbol{\mu} > 0\}$ for all $t \leq t_{s_1}$.

Finally, since $r(k)$ grows linearly in $k$, the monotonicity claim follows immediately. This proves the proposition. $\qquad\square$

### C.4 PROOF OF THEOREM 3.5

**Theorem C.7** (CFG yields stronger within-mode contraction for small $\sigma$). *Consider a class-conditional Gaussian mixture with component $k$ having covariance $\sigma^2 I_d$. Assume $\sigma^2 < 1$, and the mixture is sufficiently well-separated so that on a ball $\mathbb{B}_k(t, r) := \mathbb{B}((1 - t)\boldsymbol{\mu}_k, r)$ one has $w_k(t, \boldsymbol{x}) \geq 1 - \varepsilon$ and $\sum_j \|\nabla w_j(t, \boldsymbol{x})\| \leq C_{\mathrm{resp}}$ uniformly for all $t \in [0, t_{s_3}), \, \boldsymbol{x} \in \mathbb{B}_k(t, r)$, with $\varepsilon > 0$ arbitrarily small by taking the separation large enough and $r, t_{s_3}$ small enough. Let $\boldsymbol{x}_t^{(\mathrm{CFG})}, \boldsymbol{z}_t^{(\mathrm{CFG})}$ and $\boldsymbol{x}_t^{(y)}, \boldsymbol{z}_t^{(y)}$ be the solutions of the CFG and conditional flows (same initial pair in $\mathbb{B}_k(t_{s_3}, r)$ at time $t_{s_3}$). Then for any guidance $\omega > 1$ there exist $t_{s_3} \in (0, 1)$ and $r > 0$ (depending on the mixture separation and $\sigma$) such that for all $t \in [0, t_{s_3})$,*

$$\big\|\boldsymbol{x}_t^{(\mathrm{CFG})} - \boldsymbol{z}_t^{(\mathrm{CFG})}\big\| < \big\|\boldsymbol{x}_t^{(y)} - \boldsymbol{z}_t^{(y)}\big\|.$$

**Proof Sketch.** When the dynamics are predominantly driven by a single mode $\boldsymbol{\mu}_k$, the unconditional probability–flow ODE reads

$$\dot{\boldsymbol{x}}_t \;=\; -\frac{1}{t}\big(\hat{\boldsymbol{x}}_{0|t}(\boldsymbol{x}_t) - \boldsymbol{x}_t\big),$$

where $\hat{\boldsymbol{x}}_{0|t}(\boldsymbol{x}_t)$ is an affine combination of 0 and $\boldsymbol{x}_t$. For the difference between two trajectories, we have

$$\frac{\mathrm{d}}{\mathrm{d}t}\big(\boldsymbol{x}_t^{(\mathrm{CFG})} - \boldsymbol{z}_t^{(\mathrm{CFG})}\big) = \frac{\mathrm{d}}{\mathrm{d}t}\big(\boldsymbol{x}_t^{(y)} - \boldsymbol{z}_t^{(y)}\big) - \frac{\omega}{t}\big(\hat{\boldsymbol{x}}_{0|t}^{(y)}(\boldsymbol{x}_t) - \hat{\boldsymbol{x}}_{0|t}(\boldsymbol{x}_t) - \hat{\boldsymbol{z}}_{0|t}^{(y)}(\boldsymbol{z}_t) + \hat{\boldsymbol{z}}_{0|t}(\boldsymbol{z}_t)\big).$$

Since the samples are almost entirely governed by the same mode $\boldsymbol{\mu}_k$, we can approximate

$$\hat{\boldsymbol{x}}_{0|t}^{(y)}(\boldsymbol{x}_t) \;\approx\; \alpha_t \boldsymbol{\mu}_k + \beta_t \boldsymbol{x}_t, \qquad \hat{\boldsymbol{x}}_{0|t}(\boldsymbol{x}_t) \;\approx\; \tilde{\beta}_t \boldsymbol{x}_t,$$

with $\beta_t < \tilde{\beta}_t$ because the conditional prior is stronger ($\sigma < 1 = \sigma^{\text{uncond}}$). Subtracting the two estimators thus leaves a residual term $-c_t \boldsymbol{x}_t$ with $c_t > 0$. Hence,

$$\left(\tfrac{\mathrm{d}}{\mathrm{d}t}(\boldsymbol{x}_t - \boldsymbol{z}_t)\right)^{(\text{CFG})} = \left(\tfrac{\mathrm{d}}{\mathrm{d}t}(\boldsymbol{x}_t - \boldsymbol{z}_t)\right)^{(y)} - \tfrac{\omega}{t}\left(-c_t(\boldsymbol{x}_t - \boldsymbol{z}_t)\right).$$

Consequently,

$$\left(\tfrac{\mathrm{d}}{\mathrm{d}t}\|\boldsymbol{x}_t - \boldsymbol{z}_t\|\right)^{(\text{CFG})} > \left(\tfrac{\mathrm{d}}{\mathrm{d}t}\|\boldsymbol{x}_t - \boldsymbol{z}_t\|\right)^{(y)},$$

which establishes that CFG induces stronger contraction of pairwise distances within the same mode.

*Proof.* Fix $k \in \{1, 2\}$ and work inside $\mathbb{B}_k(t, r) := \mathbb{B}((1-t)\mu_k, r)$ for $t \in [0, t_{s_3})$. For any two solutions of the same flow, let $\Delta_t^\bullet := x_t^\bullet - z_t^\bullet$ ($\bullet \in \{(y), (\text{CFG})\}$).

*(A) Explicit posterior differences.* For a single Gaussian $\mathcal{N}(\mu_k, \sigma^2 \mathbf{I})$ we have the closed form

$$\hat{\boldsymbol{x}}_{0|t}^{(k)}(x) = \alpha_t \boldsymbol{x} + \beta_t \boldsymbol{\mu}_k, \quad \alpha_t = \frac{\sigma^2(1-t)}{t^2 + (1-t)^2\sigma^2}, \quad \beta_t = \frac{t^2}{t^2 + (1-t)^2\sigma^2},$$

and for the unconditional prior $\mathcal{N}(0, \mathbf{I})$, $\hat{\boldsymbol{x}}_{0|t}(\boldsymbol{x}) = \gamma_t \boldsymbol{x}$ with $\gamma_t = \frac{1-t}{t^2 + (1-t)^2}$. In the mixture model,

$$\hat{\boldsymbol{x}}_{0|t}^{(y)}(\boldsymbol{x}) = \sum_j w_j(t, \boldsymbol{x}) \hat{x}_{0|t}^{(j)}(\boldsymbol{x}) = \alpha_t \boldsymbol{x} + \beta_t \sum_j w_j(t, \boldsymbol{x})\boldsymbol{\mu}_j.$$

Hence, for any $\boldsymbol{x}, \boldsymbol{z}$ and $\Delta := \boldsymbol{x} - \boldsymbol{z}$,

$$\hat{\boldsymbol{x}}_{0|t}^{(y)}(\boldsymbol{x}) - \hat{x}_{0|t}^{(y)}(\boldsymbol{z}) = \alpha_t \Delta + \beta_t \boldsymbol{G}_t(\boldsymbol{x}, \boldsymbol{z})\Delta, \quad \boldsymbol{G}_t(\boldsymbol{x}, \boldsymbol{z}) := \sum_j \boldsymbol{\mu}_j \int_0^1 (\nabla w_j)(\boldsymbol{z} + s\Delta)^\top \, \mathrm{d}s. \quad (18)$$

*(B) Explicit ODE for $\Delta_t^\bullet$.* Using $\dot{\boldsymbol{x}}^{(y)}(t, \boldsymbol{x}) = -(\hat{\boldsymbol{x}}_{0|t}^{(y)}(\boldsymbol{x}) - \boldsymbol{x})/t$ and $\dot{\boldsymbol{x}}^{(\text{CFG})}(t, \boldsymbol{x}) = -(\hat{\boldsymbol{x}}_{0|t}^{(\text{CFG})}(\boldsymbol{x}) - \boldsymbol{x})/t$ with $\hat{\boldsymbol{x}}_{0|t}^{(\text{CFG})} = (1-\omega)\hat{\boldsymbol{x}}_{0|t} + \omega\hat{\boldsymbol{x}}_{0|t}^{(y)}$, combining with equation 18 gives

$$\dot{\Delta}_t^{(y)} = -\left(\tfrac{\alpha_t - 1}{t}\mathbf{I} + \tfrac{\beta_t}{t}\boldsymbol{G}_t\right)\Delta_t^{(y)}, \qquad \dot{\Delta}_t^{(\text{CFG})} = -\left(\tfrac{(1-\omega)\gamma_t + \omega\alpha_t - 1}{t}\mathbf{I} + \omega\tfrac{\beta_t}{t}\boldsymbol{G}_t\right)\Delta_t^{(\text{CFG})}. \quad (19)$$

*(C) Two-sided differential inequalities for $\log\|\Delta_t^\bullet\|$.* Let $M := \sup\limits_{t \in [0, t_{s_3}), \, x, z \in \mathbb{B}_k(t, r)} \|\boldsymbol{G}_t(x, z)\|$. Then from equation 19 and Cauchy–Schwarz,

$$a_\bullet(t) - |b_\bullet(t)| \, M \leq -\frac{\mathrm{d}}{\mathrm{d}t}\log\|\Delta_t^\bullet\| \leq a_\bullet(t) + |b_\bullet(t)| \, M, \tag{20}$$

$$\begin{cases} a_{(y)} = \dfrac{\alpha_t - 1}{t}, & b_{(y)} = \dfrac{\beta_t}{t}, \\ a_{(\text{CFG})} = \dfrac{(1-\omega)\gamma_t + \omega\alpha_t - 1}{t}, & b_{(\text{CFG})} = \omega\dfrac{\beta_t}{t}. \end{cases} \tag{21}$$

*(D) Pointwise gap.* Subtract the upper bound for (CFG) from the lower bound for $(y)$:

$$\left(a_{(y)} - b_{(y)}M\right) - \left(a_{(\text{CFG})} + b_{(\text{CFG})}M\right) = (\omega - 1)\frac{\gamma_t - \alpha_t}{t} - (1+\omega)\frac{\beta_t}{t}M.$$

Using the small-$t$ expansions

$$\gamma_t - \alpha_t = t^2(\sigma^{-2} - 1) + \mathcal{O}(t^3), \qquad \frac{\beta_t}{t} = \frac{t}{\sigma^2} + \mathcal{O}(t^2),$$

we find

$$\left(a_{(y)} - b_{(y)}M\right) - \left(a_{(\text{CFG})} + b_{(\text{CFG})}M\right) = t\left((\omega-1)(\sigma^{-2} - 1) - (1+\omega)\tfrac{M}{\sigma^2} + \mathcal{O}(t)\right).$$

Therefore, if $\sigma^2 < 1$ and $M$ is small enough so that

$$(\omega - 1)(\sigma^{-2} - 1) > (1+\omega)\tfrac{M}{\sigma^2},$$

then there exist $t_{s_3} \in (0, 1)$ and $c_0 > 0$ with

$$a_{(\text{CFG})}(t) + b_{(\text{CFG})}(t)M \; \leq \; a_{(y)}(t) - b_{(y)}(t)M \; - \; c_0 t, \qquad \forall\, t \in [0, t_{s_3}). \tag{22}$$

*(E) Concluding the comparison.* Combining equation 21 and equation 22,

$$\frac{\mathrm{d}}{\mathrm{d}t}\Big( \log \|\Delta_t^{(\text{CFG})}\| - \log \|\Delta_t^{(y)}\| \Big) \; \geq \; c_0\, t \; > \; 0 \quad \text{for all } t \in [0, t_{s_3}).$$

Since the two distances agree at $t_{s_3}$ (same initialization), integrating backward in time yields $\|\Delta_t^{(\text{CFG})}\| < \|\Delta_t^{(y)}\|$ for all $t \in [0, t_{s_3})$. $\qquad\square$

## D    DETAILMENT OF EXPERIMENT

In the experiments evaluating the impact of early high guidance (with $N = 50$ NFEs), we set the low guidance weight to 3 and the high guidance weight to 9. In the *early-high* strategy, the weight is switched from low to high after 20% of the iterations, whereas in the *late-high* strategy, the adjustment is made in the opposite direction.

In the experiments evaluating the impact of late high guidance (also with $N = 50$ NFEs), we start from the same noise initialization and inject an additional perturbation drawn from $N(0, 0.04^2)$ at 20% of the iterations. For the constant-low schedule, the guidance weight is fixed at 3 throughout. For the late-high schedule, to ensure fairness, the weight is set to 3 during the first 20% of the iterations, reduced to 1 between 20% and 60%, and increased to 5 from 60% to 100%.

The experiments were conducted with Stable Diffusion v3.5 on the COCO 2017 validation set (5,000 captioned images). For each configuration, we generated 5,000 images at $1024 \times 1024$ resolution and evaluated them using FID, CLIP, ImageReward, and saturation, covering both quality and diversity. All runs used NVIDIA A100-SXM4-40GB GPUs with `PyTorch` and Hugging Face `Diffusers`.

We compared four settings: (i) vanilla-CFG with constant guidance $\omega$; (ii) time-varying CFG (TV-CFG), which weakens guidance in early/late steps and strengthens it mid-way; (iii) $\beta$-CFG (Malarz et al., 2025), which utilizes the probability density function (PDF) of a Beta distribution as the weight schedule; and (iv) interval-CFG (Kynkäänniemi et al., 2024), which executes CFG only within a specific interval.

For TV-CFG, let $N$ be the number of NFEs, $\{t_n\}_{n=0}^{N-1}$ the timesteps, and $\{\omega_{t_n}\}$ the scales. The schedule is

$$\omega_{t_n} = \begin{cases} A\Big( \frac{2s}{\lceil N/2 \rceil} n + \omega - s \Big), & n \leq \lceil N/2 \rceil, \\ A\Big( \frac{2s}{\lceil N/2 \rceil} (N - n) + \omega - s \Big), & n > \lceil N/2 \rceil, \end{cases}$$

with $s = \omega - 1$. The factor $A$ normalizes the average scale to match the baseline:

$$\sum_{n=0}^{N-1} \omega_{t_n} (t_n - t_{n+1}) = \omega, \quad t_N = 0,$$

analogous to $\int_0^1 \omega_t\, \mathrm{d}t = \omega$. For interval-CFG and $\beta$-CFG, the values of the guidance scale within the CFG interval are also determined by normalization, and the relevant hyperparameters are set according to the recommendations in the original paper. Figure 5 visualizes the schedules.

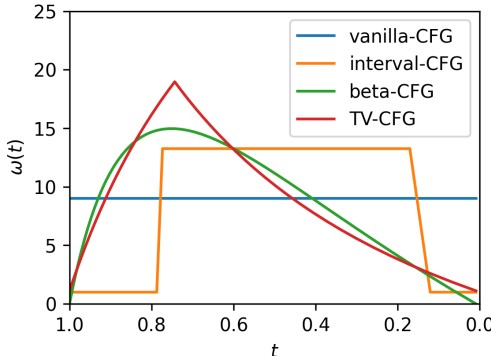

Figure 5: Vanilla-CFG, interval-CFG, $\beta$-CFG and TV-CFG guidance scale settings at $\omega = 9$.

Diversity is evaluated by sampling 1,000 prompts from the COCO dataset. For each prompt, 16 images are generated using different random seeds. LPIPS features are extracted for all generated images, and diversity is quantified as the mean squared pairwise distance between feature representations. Larger values correspond to higher sample diversity.

# E   ABLATION STUDY ON PEAK TIMING

In this section, we investigate the sensitivity of our method to the peak location of the guidance schedule. Our theoretical framework describes a three-stage dynamic: early direction shift, intermediate mode separation, and late contraction. The symmetric TV-CFG schedule used in our main experiments serves as a minimal, theory-aligned instantiation. To demonstrate that the effectiveness relies on the general stage-wise mechanism rather than fine-grained hyperparameter tuning, we conduct an ablation study varying the peak timing.

We adjust the peak position of the schedule to occur at 20%, 40%, 60%, and 80% of the total sampling steps, fixing the total steps at NFE = 10 and the guidance scale at $\omega = 7$. We evaluate the performance across CLIP, IR, FID, saturation, and Diversity metrics.

Table 3: Ablation study on the sensitivity of the peak timing location. Experiments were conducted with NFE = 10 and $\omega = 7$. The results demonstrate robustness when the peak lies within the intermediate regime.

| Peak Timings | CLIP ↑ | IR ↑ | FID ↓ | Saturation | Diversity ↑ |
|---|---|---|---|---|---|
| 20% | 0.317 | 0.752 | 31.669 | 0.401 | 1.2214 |
| 40% | **0.319** | **0.940** | 28.770 | 0.279 | **1.2309** |
| 60% | **0.319** | 0.935 | 27.722 | 0.229 | 1.1963 |
| 80% | **0.319** | 0.896 | **27.092** | 0.201 | 1.1622 |

The quantitative results are presented in Table 3. We observe the following:

- **Stability in the Intermediate Regime:** When the peak is located within the intermediate stage (40%–60%), the performance remains stable and optimal across most metrics. Notably, CLIP scores saturate at their highest value (**0.319**), and IR reaches its peak at 40%.

- **Impact of Extreme Timings:** Setting the peak too early (20%) or too late (80%) leads to expected degradation in specific areas. An early peak (20%) results in lower IR and higher FID, suggesting that applying strong guidance too soon hinders the quality of the samples. Conversely, a late peak (80%) yields the lowest Diversity (1.1622) and Saturation, consistent with the theoretical prediction that delayed guidance restricts the available generation space.

These findings confirm that the primary factor for success is the "low → high → low" structural shape predicted by our three-stage theory. The method is robust to the exact peak location, provided it resides within the mode-separation stage. This supports our conclusion that the stage-wise mechanism, rather than precise hyperparameter selection, governs the behavior of guided sampling.

## F  ADDITIONAL VISULIZATION RESULTS

Figures6, and 7 present results under different prompts and scheduling strategies, further illustrating how strong early guidance undermines global diversity (NFE=50, $\omega = 9$). Figures 8, 9, 10, and 11 demonstrate the effect of the proposed time-varying schedule on the generated images (NFE=20, $\omega = 9$).

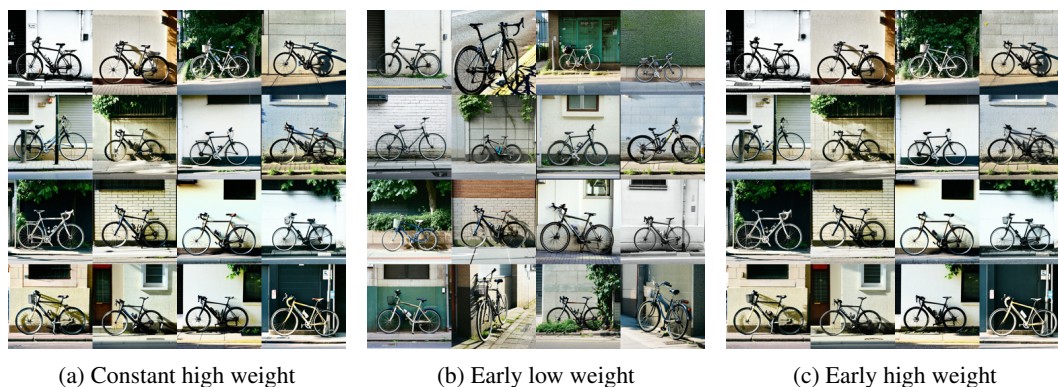

(a) Constant high weight          (b) Early low weight          (c) Early high weight

Figure 6: Comparison of guidance schedules on the prompt *"a bike parked on a side walk."*. (a) Constant high weight and (c) Early high weight both exhibit directional collapse, with nearly all bicycles oriented in the same side-facing position.

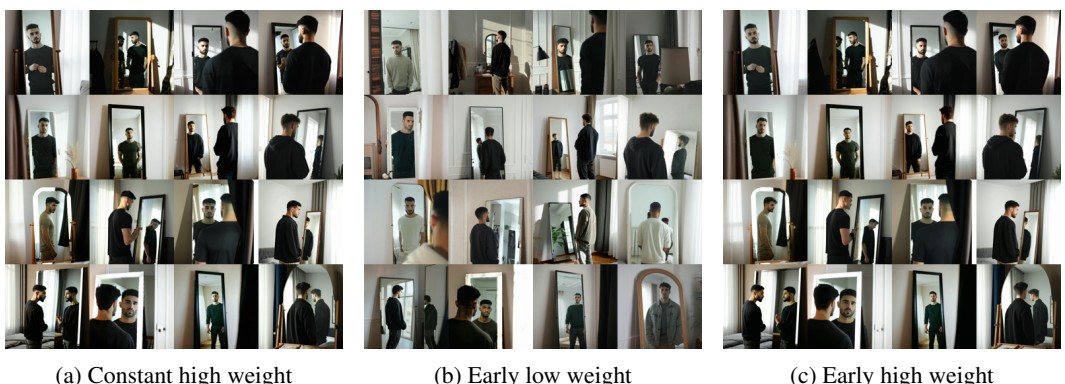

(a) Constant high weight          (b) Early low weight          (c) Early high weight

Figure 7: Comparison of guidance schedules on the prompt *"A man standing in front of a mirror in a room."*. (a) Constant high weight and (c) Early high weight both reduce diversity, with most samples converging to nearly identical settings: a man in a dark shirt facing a tall rectangular mirror.

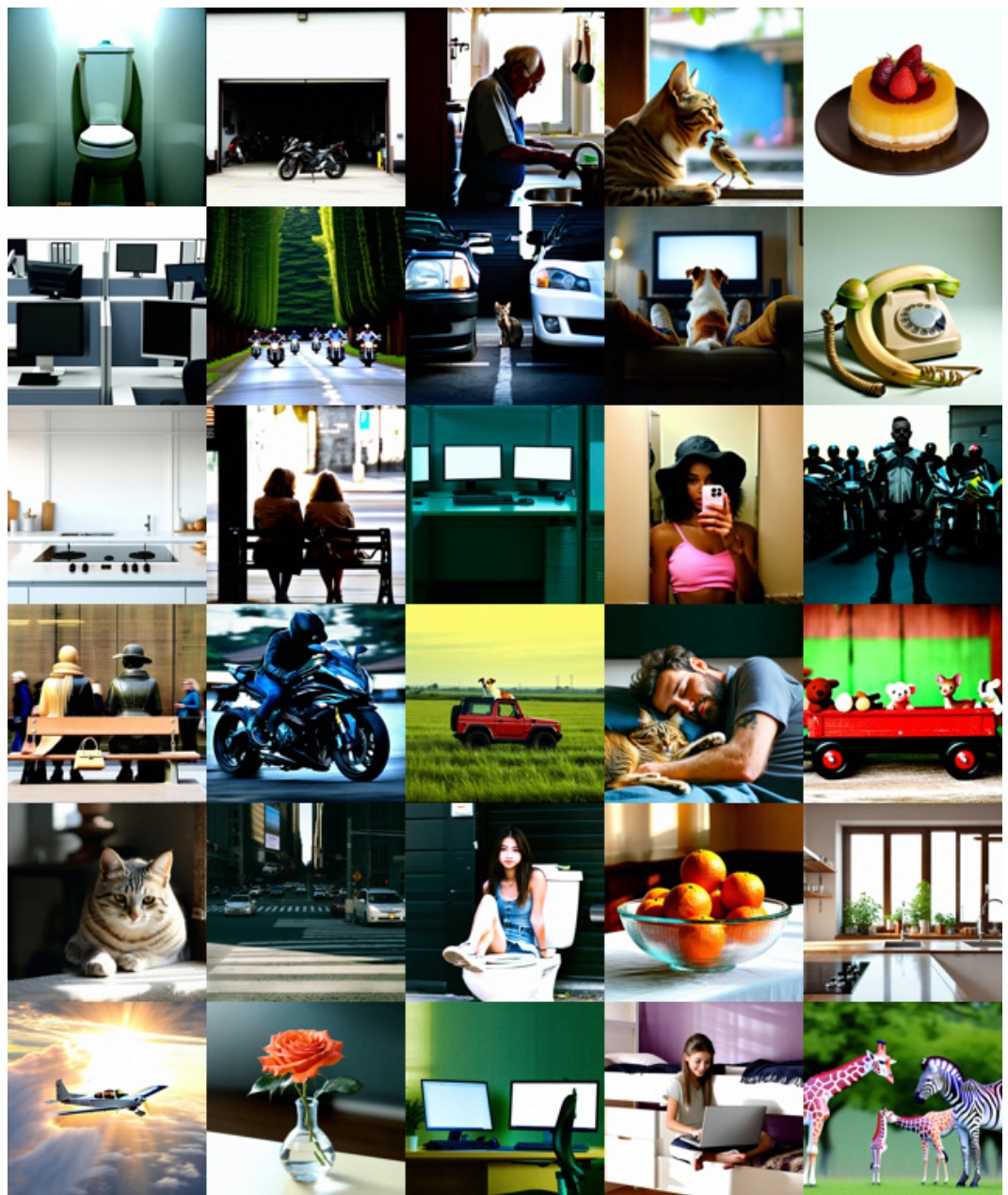

Figure 8: Generated samples using CFG.

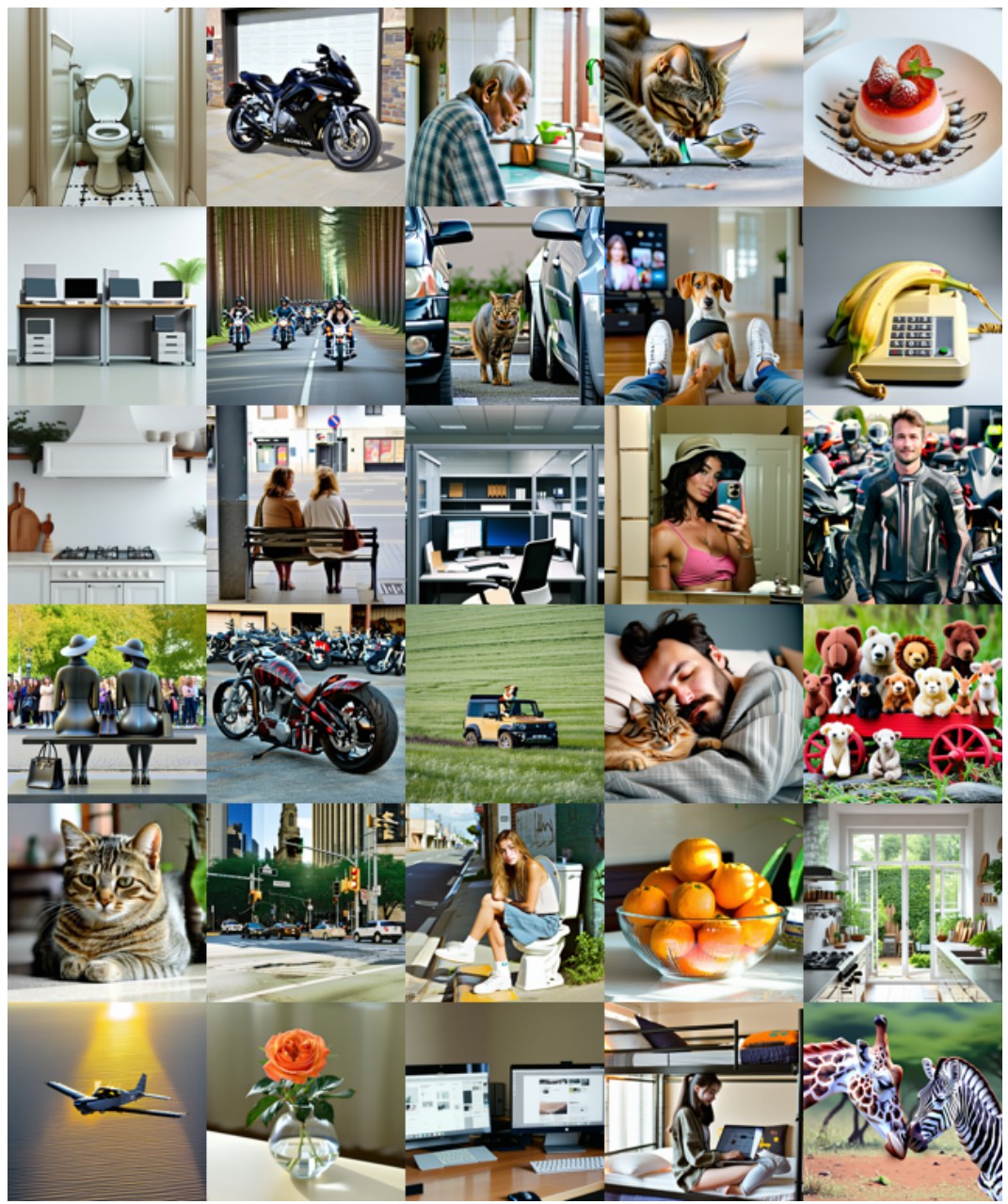

Figure 9: Generated samples using TV-CFG.

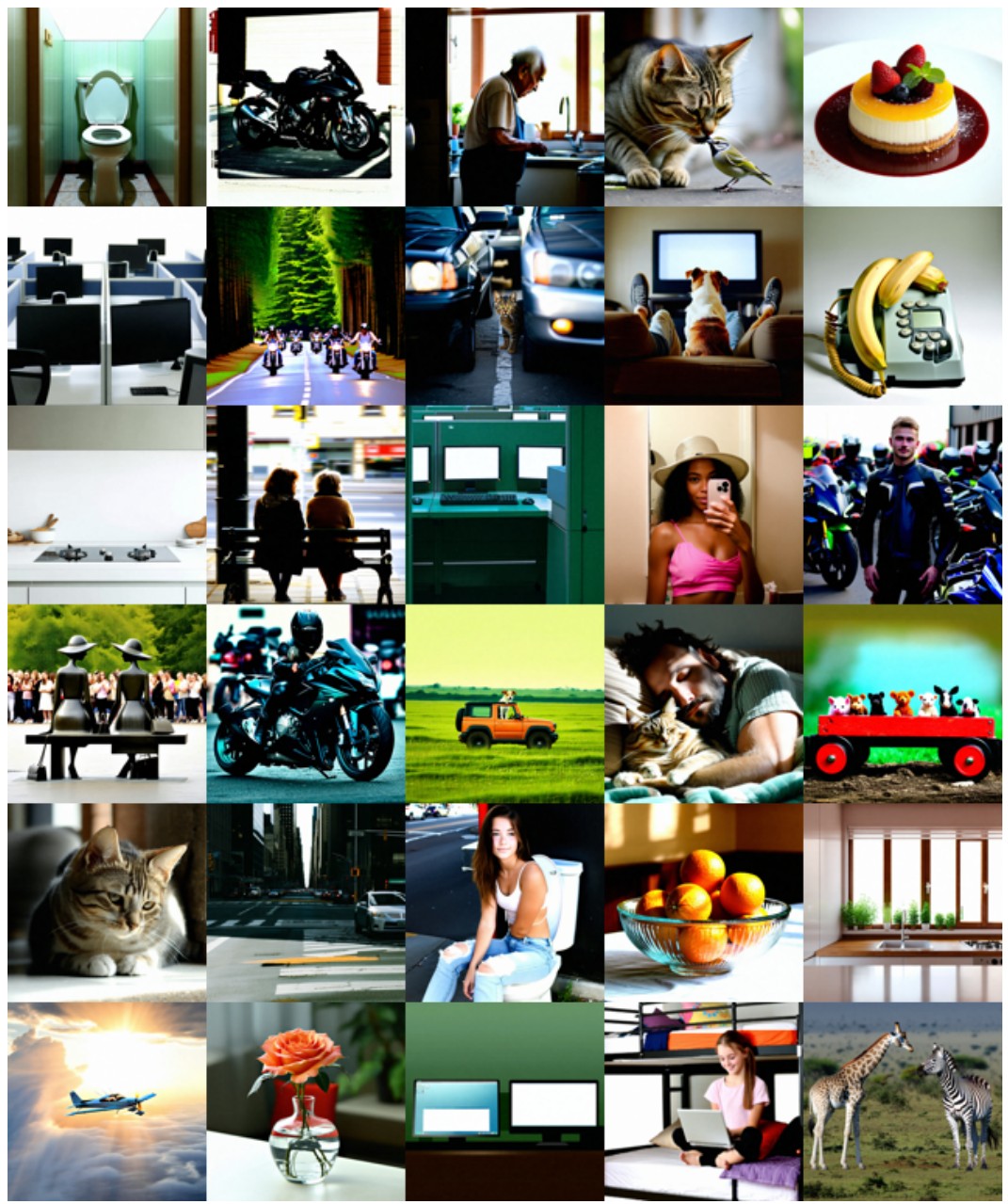

Figure 10: Generated samples using APG.

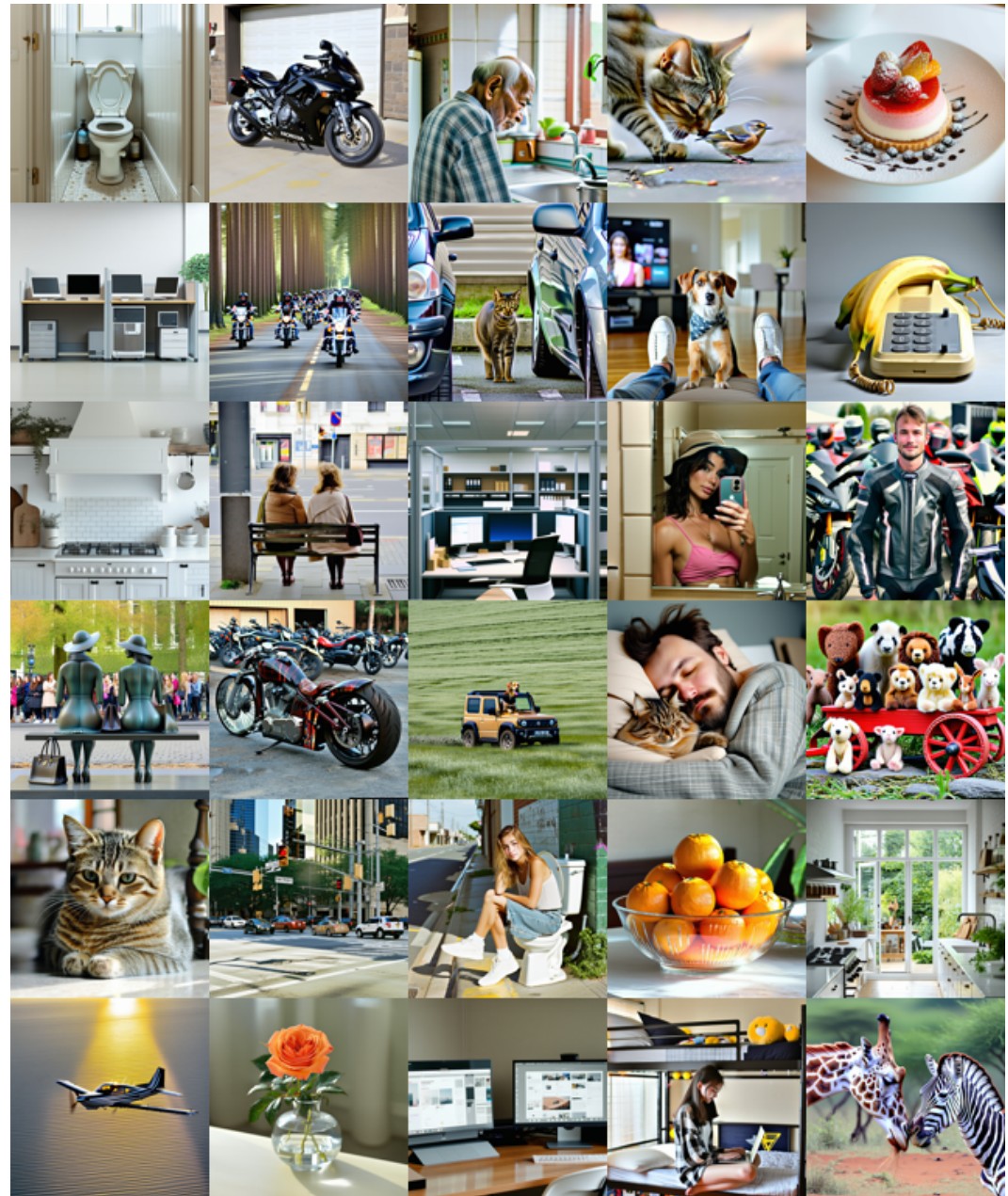

Figure 11: Generated samples using TV-APG.

