# OpenReview forum: "Stage-wise Dynamics of Classifier-Free Guidance in Diffusion Models"
_ICLR.cc/2026/Conference — ICLR 2026 Poster_

### Official Review · Reviewer_gN79 · 2025-10-23

**Soundness:** 3
**Presentation:** 4
**Contribution:** 3
**Rating:** 8
**Confidence:** 4

**Summary:**

The main contribution of this paper is the description of three distinct phases in the application of Classifier-Free Guidance (CFG). First, in a "strong noise" phase, CFG causes an *acceleration and direction shift*, pushing the distribution toward the global, class-weighted mean of the dataset. This essentially acts as an amplification process that inflates the norms of the trajectories. The authors claim this creates an initialization effect that biases the process toward lower-diversity samples.

The second phase is *mode separation*, where trajectories begin fanning out toward the different modes in the data distribution as noise decreases, settling into their basins of attraction. The issue here, as the authors argue, is that the initial stage has already biased the results, positioning trajectories to favor certain modes over others.

Finally, the third phase is *concentration*, which occurs in the low-noise regime. Here, the dynamics are dominated by contraction, which suppresses intra-class variability. This results in samples that are sharper but less diverse.

Additional contributions of the paper include empirical validation of the theory and a time-varying guidance protocol for diffusion sampling.

**Strengths:**

Overall, my impression of the paper is quite positive.

* The paper is well-written and well-presented. The illustrations, especially in Figure 1, are very helpful for understanding the different stages the authors discuss.
* A significant contribution of this work is that it lends some theoretical muscle to analyzing CFG. CFG is a technique we all use, but it's essentially a hack that works without a tremendous amount of theory to motivate it. It is valuable to see an analysis from a dynamical perspective, especially as the field moves more toward ODE representations of the diffusion process. This is a genuine contribution to the literature and provides food for thought for additional theoretical development.
* The theoretical analysis itself, which models the data as a Gaussian mixture, is general enough to provide useful insight, seems sound, and provides a good foundation for the claims. However, see the caveats introduced below.

**Weaknesses:**

* I occasionally struggled to keep track of some of the theoretical results in the theorems. In some cases, the meaning or interpretation of certain variables wasn't entirely clear. For instance, in Proposition 3.4, a parameter $k$ is introduced that determines a radius. It's later stated that $k$ increases with the CFG weight $\omega$ (line 273), but it was not completely clear to me why that is the case. It was also not clear to me what "mild assumptions" were being made in some of the theorems. To the extent possible without introducing too much clutter, making the theoretical statements more self-contained would be helpful.
* In the experimental results, the authors' proposed method is referred to as "TV." While it can be inferred that TV means “time-varying,” the paper does not explicitly define what "TV" stands for immediately before abbreviating it (unless I missed it). So when looking at the experimental tables, I wasn't sure at first if the "TV" results belonged to the authors or someone else.
* The authors refer to the difference between "global diversity," which is negatively affected by strong early guidance, and "fine-grained diversity," which is reduced by strong late guidance. While I have an intuitive sense of what the authors mean by this terminology, I would like to see a more precise definition of these two concepts.
* The paper would benefit from discussing its findings in the context of some other relevant work:
1) The CADS method [1] uses a condition-annealing strategy during inference, where noise is added to the conditioning signal early on in sampling and is then gradually reduced to reveal an unperturbed condition. It would be interesting to see the authors reconcile their theoretical results with that technique. I would also refer to the "fine-grained detail" argument the authors make to see how it reconciles with the results reported in the CADS paper.
2) Two recent papers [2, 3] describe the *biphasic* nature of diffusion sampling, which are distinguished by the relative influence of the conditioning signal. It would be interesting to have the authors discuss their "triphasic" findings in the context of this other work.
* While I think the authors' modeling choices allow for insight into the more general conditional diffusion sampling problem, the theoretical results still rely on what is essentially a toy setting. To the authors' credit, their empirical results operate on real in-the-wild diffusion models. However, a more frank acknowledgement and discussion of the limitations of the theoretical results is needed.
* I noticed very few (but nonzero) typos in the paper (e.g. an extra period in the caption for Figure 2, using *assumption* where *assumptions* is intended in one theorem statement, etc.). One more reading pass, especially from a fresh-eyed reader, should help eliminate them.

[1] Sadat et al. (2023). CADS: Unleashing the Diversity of Diffusion Models through Condition-Annealed Sampling

[2] Balaji et al. (2022). eDiff-I: Text-to-Image Diffusion Models with an Ensemble of Expert Denoisers

[3] Liu et al. (2024). Faster Diffusion: Rethinking the Role of Temporal Attention in Diffusion Models

**Questions:**

* Given the use of $\omega > 1$ in Theorem 3.2, am I correct in the interpretation that trajectories are directed toward the class-weighted mean scaled *exactly* by the CFG scale? If that's the case, it seems like a result that should be more explicitly called out, especially given the large CFG scales often used in practice. (And if that's not the conclusion here, it would be a confusing reuse of notation.)
* The authors’ theoretical analysis relies on the countable, discrete components of a GMM. This setting is well-suited to class-conditional models but less obviously applies to uncountable, continuous conditions (e.g. CLIP embeddings). What is the authors' perspective on how to extend their analysis to the continuous-condition domain?
* Please also address the issues described in the "Weaknesses" section.

**Details Of Ethics Concerns:**

No concerns.

---

> ### Author Response · Authors · 2025-11-13
> **part 1 of the rebuttal**
>
> We deeply appreciate your insightful and constructive feedback.
> We are encouraged that you recognized the central contribution of our work as providing a principled theoretical framework for understanding the dynamics of Classifier-Free Guidance (CFG), with the experimental section serving primarily to validate our theoretical predictions and to confirm the effectiveness of the time-varying guidance schedule. We also sincerely thank you for carefully reading our manuscript and pointing out minor typos and formatting issues. Below, we address each of your comments in detail.
>
> ------
>
> ## Response to Weaknesses
>
> ### Clarification of Proposition 3.4 and the Role of the CFG Weight
>
> We will clarify more explicitly the relationship between **k** in Proposition 3.4 and the guidance weight **ω**.
>  Proposition 3.4 shows that, at an intermediate stage, points within the radius **r(k)** around **kμ̄** remain in the basin of the dominant mode.
>  According to Theorem 3.2, a larger **ω** causes early-stage trajectories to be closer to **ωμ̄**,
>  corresponding to a larger effective **k** and thus expanding the region dominated by the stronger mode.
>  This interaction between the early-stage directional bias (Theorem 3.2) and the mid-stage dynamics (Proposition 3.4)
>  explains why trajectories increasingly concentrate on dominant modes, leading to reduced global diversity.
>
> ------
>
> ### Clarification of the “Mild Condition”
>
> We appreciate the reviewer’s suggestion on improving clarity.
>  The mild condition for Theorem 3.3 is explained in Appendix C.2 (Remark C.5).
>  It was omitted from the theorem statement to avoid excessive length, as it depends on the definitions of **S_t** and **g_t**.
>  In short, the sufficient condition ensuring the existence of **S_t** is intuitive:
>  the dominant and weaker modes need only be sufficiently separated.
>  We will make this clearer in the revised version to enhance readability.
>
> ------
>
> ### Definition of “TV”
>
> The reviewer’s understanding is correct.
>  We will explicitly define “TV” as the abbreviation of **time-varying** in the revised manuscript.
>
> ------
>
> ### Definition of Global and Fine-Grained Diversity
>
> We will make the definitions of these two concepts more precise in the revision.
>  **Global diversity** refers to the coverage of different mixture components (semantic modes) in the conditional distribution,
>  while **fine-grained diversity** quantifies intra-mode variability among samples belonging to the same class.
>  This distinction provides a clearer foundation for understanding the separate effects of early and late guidance.
>
> ------
>
> ### Discussion with Related Practical Works
>
> We thank the reviewer for highlighting three representative papers—CADS [1], eDiff-I [2], and Faster Diffusion [3].
>  We will cite and discuss them in the revised version.
>  Importantly, these studies are grounded in **empirical observations** rather than formal theoretical analyses,
>  and thus provide valuable practical evidence supporting our theoretical understanding.
>  Specifically, CADS introduces a condition-annealing strategy that effectively weakens early guidance and strengthens it later,
>  which is consistent with our observation that using weaker guidance in early stages helps preserve global diversity.
>  eDiff-I reports that early diffusion steps determine the global semantic layout of an image,
>  whereas later steps refine its fine-grained visual details.
>  Similarly, Faster Diffusion finds that, when using a U-Net architecture,
>  encoder outputs exhibit weak temporal variation while decoder outputs change more significantly over time.
>  Both observations imply the same underlying phenomenon:
>  the **early diffusion process establishes coarse-grained semantics**,
>  while the **late diffusion process determines fine-grained structures**.
>  This biphasic nature of diffusion sampling resonates strongly with our theoretical and empirical analysis of CFG,
>  where we demonstrate that **early CFG primarily affects global diversity**,
>  while **late CFG governs fine-grained diversity**.
>
> ------
>
> ### On the Idealized Theoretical Assumptions
>
> We appreciate the reviewer’s attention to the assumptions underlying our analysis.
>  As in most theoretical studies on CFG, our analysis adopts an idealized distributional setting for analytical tractability.
>  This simplification is a methodological necessity rather than a neglect of practical complexity.
>  Compared with prior theoretical works, our assumptions are closer to the real behavior of diffusion models,
>  and our experiments on large-scale pretrained models provide strong empirical validation of our theoretical predictions.
>  We will include a more explicit discussion of these limitations in the revised version
>  to clarify the scope and applicability of our conclusions.
>
> ------
>
> ### Minor Typos and Formatting
>
> We appreciate the reviewer’s careful reading and will correct all minor typographical and formatting issues.

---

> > ### Author Response · Authors · 2025-11-13
> > **part 2 of the rebuttal**
> >
> > ## Response to Questions
> >
> > ### Clarification of the Use of ω in Theorem 3.2
> >
> > The reviewer’s interpretation is correct.
> >  We consistently use **ω** to denote the CFG scale throughout the paper
> >  and will explicitly clarify this in Theorem 3.2 to avoid any potential ambiguity.
> >
> > ------
> >
> > ### Extension to Continuous Conditional Domains
> >
> > Although our theoretical analysis is developed under countable Gaussian-mixture conditionals,
> >  the early-stage result (Theorem 3.2)—that under strong noise, trajectories are biased toward the scaled conditional mean
> >  **ω E[x|y]**—extends naturally to more general **continuous** condition spaces (e.g., CLIP embeddings),
> >  since it does not depend on the mixture structure (Remark C.3).
> >  Moreover, when semantic modes are approximately orthogonal in high-dimensional settings,
> >  the weighted mean **μ̄** tends to align more closely with the dominant mode,
> >  yielding a Proposition 3.4–like absorption effect in which the early-stage displacement increases the likelihood of entering dominant basins.
> >  Consequently, the coupling of Stage I directional bias and Stage II mode-separation dynamics
> >  continues to produce a loss of **global diversity** (i.e., weaker modes are undersampled) even in continuous domains.
> >  In contrast, a precise theoretical characterization of the late-stage **concentration effect**
> >  would require additional assumptions about the **intra-mode geometry**.
> >
> > ------
> >
> > **References**
> >
> > [1] Sadat et al. (2023). *CADS: Unleashing the Diversity of Diffusion Models through Condition-Annealed Sampling.*
> >
> > [2] Balaji et al. (2022). *eDiff-I: Text-to-Image Diffusion Models with an Ensemble of Expert Denoisers.*
> >
> > [3] Liu et al. (2024). *Faster Diffusion: Rethinking the Role of Temporal Attention in Diffusion Models.*
> >
> > ------
> >
> > Regarding the revisions we mentioned in the rebuttal, we will notify you immediately once the updated version of the paper has been completed and uploaded.

---

> > > ### Comment · Reviewer_gN79 · 2025-11-20
> > > **Response to Authors**
> > >
> > > I thank the authors for their detailed response to my review. I am satisfied to keep my score where it is.

---

> > > > ### Author Response · Authors · 2025-11-22
> > > >
> > > > Thank you for the update. We appreciate your careful evaluation and your time.

---

### Official Review · Reviewer_v1Re · 2025-10-25

**Soundness:** 3
**Presentation:** 3
**Contribution:** 2
**Rating:** 4
**Confidence:** 3

**Summary:**

This paper presents a theoretical analysis of Classifier-Free Guidance (CFG) in diffusion models under multimodal conditional distributions. The authors identify three distinct stages in the sampling dynamics: Direction Shift (early), Mode Separation (middle), and Concentration (late), and explain how CFG leads to diversity loss through these stages. They validate their theory empirically and propose a time-varying guidance schedule.

**Strengths:**

1.  The paper provides a systematic analysis of CFG under multimodal conditional distributions, moving beyond restrictive unimodal assumptions prevalent in prior work.

2. The paper successfully explains the long-standing empirical observation that strong CFG improves semantic alignment but reduces diversity.

3. The experiments on Stable Diffusion v3.5 with COCO validation set appropriately test the theoretical assessments.

**Weaknesses:**

1.  In Related Work, the treatment of closely related concurrent papers, say, both the  theoretical or algorithmic perspectives on conditional sampling and CFG in diffusion is incomplete.

2. While experiments are run on a modern diffusion model and COCO captions, the diversity of datasets and generative tasks is limited. For example, all primary experiments use a single text-to-image model/configuration (Stable Diffusion v3.5 with 5,000 COCO images).  Discussion on how these findings generalize to other modalities (e.g., audio, video) or other conditional types (e.g., class or style transfer) are missing.

3. Although time-varying and interval guidance competitors are implemented (and TV-CFG is compared), the 'interval' baseline is only briefly described, and no discussion is given to recently proposed, possibly stronger or more customized adaptive schedules. Moreover, the ablation or sensitivity analysis on schedule shapes, peak timings, or trade-offs beyond the one setting is insufficient.

4. The central theoretical perspective is attributed as the 'first systematic analysis' of CFG under multimodal conditionals (see Abstract, Page 2, and Conclusion). However, several recent works—some cited, some missing[1][2][3], have begun to analyze guidance dynamics in GMMs or with alternative assumptions (Li et al. ICML25/ arxiv 2025a,  Jin et al. ICML25, Wu et al. ICML24). The authors should more carefully disentangle which aspects are genuinely new (e.g., the particular stage-wise decomposition or explicit formalism) as opposed to extended or sharpened results over antecedents.

5. The theory and results emphasize the effect of guidance at various sampling budgets (notably for low NFE), yet the paper lacks comprehensive ablation over NFE and guidance range. Only a select number of NFE and $\omega$ values are tested, how does the approach performs at extremely low (NFE < 5) or much higher NFE values?


[1] W. Deng et al., *Reflected Schrödinger Bridge for Constrained Generative Modeling. 2024.*

[2] X. R. Li. et al., *A Simple Approach to Unifying Diffusion-based Conditional Generation. 2024.*

[3]  G. M. Favero, et al., *Conditional Diffusion Models are Medical Image Classifiers that Provide Explainability and Uncertainty for Free. 2025.*

**Questions:**

refer to weaknesses.

---

> ### Author Response · Authors · 2025-11-20
> **part1 of the rebuttal**
>
> Thank you very much for giving us the opportunity to further clarify the contributions of our work. We first emphasize that the primary contribution of this paper is **theoretical**, while TV-CFG is merely a **natural and minimalist time-varying schedule derived directly from our theory**.
>  The core of our work is that, for the first time, **previous heuristic or empirically motivated time-varying schedules** [1,2,3]—which were known to improve diversity (even though they are not all aimed at enhancing diversity) —now have a **rigorous theoretical explanation**.
>  We theoretically characterize *how* and *at which sampling stages* CFG affects diversity, and we empirically demonstrate that even the simplest “increase–then–decrease’’ schedule implied by theory consistently improves diversity in real diffusion models.
>
> 1. **The reviewer explicitly identifies the relevant references in Point 4, and we respond to them in that section.**
>
> 2. While extending the evaluation to text-to-video or text-to-audio models would indeed be beneficial, in practice, improvements to CFG sampling are typically evaluated on a **single T2I diffusion model** due to the implementation cost [1,2,4,8].
>
> 3. In the appendix, we provide the interval-based time-varying schedule and follow the hyperparameter settings exactly as recommended in the original paper. Although TV-CFG serves primarily as a natural validation and theoretical consequence of our analysis (indeed, even the simplest rise–then–fall pattern already improves diversity), it is not the focus of our paper. Nevertheless, to address the reviewer’s comment, we additionally include below a sensitivity study on NFE=10, examining how different peak timings influence performance.
>
> | Metric         | peak timings | CFG (ω=5)  |
> | -------------- | ------------ | ---------- |
> | **CLIP ↑**     | 2            | 0.317      |
> |                | 4            | **0.319**  |
> |                | 6            | **0.319**  |
> |                | 8            | **0.319**  |
> | **IR ↑**       | 2            | 0.752      |
> |                | 4            | **0.940**  |
> |                | 6            | 0.935      |
> |                | 8            | 0.896      |
> | **FID ↓**      | 2            | 31.669     |
> |                | 4            | 28.770     |
> |                | 6            | 27.722     |
> |                | 8            | **27.092** |
> | **Saturation** | 2            | 0.401      |
> |                | 4            | 0.279      |
> |                | 6            | 0.229      |
> |                | 8            | 0.201      |
>
> As seen above, while the precise peak timing induces mild quantitative variation, the overall trend remains consistent: **moderate peak locations (4–6) offer a favorable balance**, whereas overly early or late peaks are slightly less stable. This further supports our claim that the key determinant is the stage-wise mechanism itself rather than any specific handcrafted schedule.
>
> These results directly respond to the reviewer’s concern:
>
> > “Although time-varying and interval guidance competitors are implemented (and TV-CFG is compared), the ‘interval’ baseline is only briefly described, and no discussion is given to recently proposed, possibly stronger or more customized adaptive schedules. Moreover, the ablation or sensitivity analysis on schedule shapes, peak timings, or trade-offs beyond the one setting is insufficient.”
>
> Our added analysis clarifies that (i) multiple peak configurations behave consistently with the theoretical predictions, and (ii) the performance variation across schedule shapes is small relative to the stage-wise phenomena our theory identifies. Thus, while designing sophisticated schedules is an interesting empirical direction, it is orthogonal to—and not required for—the validity or significance of our theoretical contribution.

---

> ### Author Response · Authors · 2025-11-20
> **part 2 of the rebuttal**
>
> 4. Although from context it is clear that *“multimodal conditionals”* refers to the scenario **“the conditional distribution is multimodal’’**, we agree that the phrasing may cause confusion.
>    We will revise *“multimodal conditionals’’* to **“multimodal conditional distribution’’** to avoid ambiguity.
>    Our intended meaning is:
>
> > *“We provide the first theoretical analysis of CFG sampling when the **conditional distribution is multimodal**.”*
>
> Extending the conditional distribution to multimodal settings significantly increases analytical difficulty, but it leads to deeper insights into diversity.
>
> Regarding the reviewer’s comments on papers with apparently similar citations:
>
> - **[5]** studies convergence rates of unconditional diffusion models under a Gaussian mixture prior—this does **not** involve conditional sampling or CFG (Li et al., ICML’25).
> - **[6]** analyzes CFG under *much weaker assumptions* without specifying the distributional form, showing that confidence improvement holds only *on average*, not uniformly over initializations(Li et al., arxiv’25a).
> - **[7]** analyzes sampling under **unimodal conditional** Gaussian and **unconditional** Gaussian mixtures (Wu et al., ICML’24), showing that CFG increases class confidence.
> - **[8]** extends [7], analyzing norm inflation and anomalous diffusion under the **same unimodal conditional** setting (Jin et al.).
>
> Our analysis is based on **multimodal conditional distributions**, and we rigorously prove that CFG sampling exhibits **three stages**: early-stage *Direction Shift*, mid-stage *Mode Separation*, and late-stage *Concentration*.
>  A detailed discussion of these references (and others) is provided in the appendix under Related Work.
>
> Regarding the three new citations suggested by the reviewer:
>
> - **[9]** proposes methods for sampling within constrained domains. Although unrelated to CFG, we can cite it as part of theoretical works on diffusion models.
> - **[10]** adds noise to both (x) and (y) and learns a reverse denoising process; sampling then uses this joint diffusion model.
> - **[11]** applies conditional diffusion models to medical imaging, using reconstruction error for diagnosis.
>   Among these, [10] and [11] *use* CFG during sampling but do **not analyze CFG itself**. We can cite them to further emphasize the broad applicability of CFG in diffusion sampling.
>
> 5. We conducted experiments at NFE = 5, 10, 20 with guidance scales = 3, 5, 7, 9.
>    Given that experiments on large SOTA models such as SDv3.5-large are extremely costly, and compared with prior works [1,2,3,4,8], this is not a narrow setting.
>    For NFE < 5, the generated images become extremely poor; see:
>    https://ibb.co/RTpdXjXS
>
> Meanwhile, we additionally include performance at **NFE = 50** as shown below:
>
> | Metric          | Variant  | CFG (NFE=50) | APG (NFE=50) |
> | --------------- | -------- | ------------ | ------------ |
> | **CLIP ↑**      | vanilla  | **0.319**    | **0.320**    |
> |                 | interval | 0.315        | 0.316        |
> |                 | TV(Ours) | 0.317        | 0.317        |
> | **IR ↑**        | vanilla  | **1.057**    | 1.076        |
> |                 | interval | 0.875        | 0.875        |
> |                 | TV(Ours) | 1.054        | **1.079**    |
> | **FID ↓**       | vanilla  | 75.050       | 75.166       |
> |                 | interval | **72.159**   | **72.172**   |
> |                 | TV(Ours) | 75.402       | 74.495       |
> | **Saturation**  | vanilla  | 0.349        | 0.344        |
> |                 | interval | 0.250        | 0.244        |
> |                 | TV(Ours) | 0.336        | 0.312        |
> | **Diversity ↑** | vanilla  | 1.211        | 1.209        |
> |                 | interval | 1.202        | 1.200        |
> |                 | TV(Ours) | **1.235**    | **1.237**    |
>
> Here, *Diversity* is computed by randomly selecting 1k COCO prompts, generating 16 images with 16 seeds for each prompt, computing LPIPS features, and reporting the mean squared pairwise distance. Higher values indicate greater sample diversity.
>
> We reiterate that the **main contribution** of the paper is to provide a **theoretical explanation** for diversity loss under CFG. Our experiments show that even the most straightforward “increase–then–decrease” schedule—derived directly from our theory—improves diversity in practical diffusion models. The goal is **not** to propose a new SOTA scheduling algorithm.

---

> > ### Author Response · Authors · 2025-11-20
> > **part 3 of the rebuttal**
> >
> > **reference**
> >
> > [1] Kynkäänniemi, Tuomas, et al. "Applying guidance in a limited interval improves sample and distribution quality in diffusion models." *Advances in Neural Information Processing Systems* 37 (2024): 122458-122483.
> >
> > [2] Malarz, Dawid, et al. "Classifier-free Guidance with Adaptive Scaling." *arXiv e-prints* (2025): arXiv-2502.
> >
> > [3] Wang, Xi, et al. "Analysis of Classifier-Free Guidance Weight Schedulers." *Transactions on Machine Learning Research Journal* (2024).
> >
> > [4] Chung, Hyungjin, et al. "Cfg++: Manifold-constrained classifier free guidance for diffusion models." *arXiv preprint arXiv:2406.08070* (2024).
> >
> > [5] Gen Li, Changxiao Cai, and Yuting Wei. Dimension-free convergence of diffusion models for approximate gaussian mixtures. arXiv preprint arXiv:2504.05300, 2025a.
> >
> > [6] Gen Li and Yuchen Jiao. Provable efficiency of guidance in diffusion models for general data distribution. ICML25
> >
> > [7] Wu Y, Chen M, Li Z, Wang M, Wei Y. Theoretical insights for diffusion guidance: A case study for Gaussian mixture models. ICML24
> >
> > [8] Cheng Jin, Zhenyu Xiao, Chutao Liu, and Yuantao Gu. Angle domain guidance: Latent diffusion requires rotation rather than extrapolation. ICML25
> >
> > [9] W. Deng et al., *Reflected Schrödinger Bridge for Constrained Generative Modeling. 2024.*
> >
> > [10] X. R. Li. et al., *A Simple Approach to Unifying Diffusion-based Conditional Generation. 2024.*
> >
> > [11] G. M. Favero, et al., *Conditional Diffusion Models are Medical Image Classifiers that Provide Explainability and Uncertainty for Free. 2025.*

---

### Official Review · Reviewer_2UBn · 2025-10-30

**Soundness:** 3
**Presentation:** 3
**Contribution:** 2
**Rating:** 4
**Confidence:** 4

**Summary:**

This paper theoretically analyzes the effect of CFG on sampling dynamics and shows that its influence can be divided into three stages. In the Direction Shift stage (early in sampling), guidance pushes samples toward a weighted average of class means, potentially biasing the initialization toward specific generations. In the Mode Separation stage, samples are driven toward stronger modes, reducing the diversity of generated examples. In the Concentration stage, strong guidance further decreases diversity while improving alignment. Overall, the authors conclude that early strong guidance reduces global diversity, while late strong guidance suppresses fine-grained variations. This motivates the use of a time-varying guidance schedule (TV-CFG) that peaks in the middle, and the experiments demonstrate that TV-CFG improves the quality–diversity trade-off compared to standard CFG.

**Strengths:**

* The paper studies the influence of guidance on sampling dynamics, which is an important aspect of modern diffusion models. Given the widespread use of CFG, further intuition in this domain is essential to better understand and improve its behavior. The work aims to bridge the gap between current empirical practices using CFG and the theoretical understanding of its underlying mechanisms.

* The authors validate their findings not only on toy datasets but also on real-world text-to-image models.

* The paper is well-written and easy to follow. The examples effectively illustrate each aspect of the discussion.

**Weaknesses:**

In my opinion, the main weakness of the paper is that its findings may not fully justify the novelty claim. Specifically, [1] studied the dynamics of CFG in Gaussian mixture models, showing that strong guidance improves classifier confidence (i.e., better alignment) while reducing the final entropy of the generated distribution (i.e., lower diversity). They also analyzed the influence of guidance across different stages of sampling (see Section 6 and Proposition 6.1). However, this reference is described by the authors as focusing only on unimodal distributions and “overlooking the multimodal nature of real-world tasks” (L42).

Moreover, the benefits of time-varying guidance scales have already been explored in prior work from a practical standpoint, as acknowledged by the authors. Therefore, I believe the novelty claims require stronger justification to warrant a separate publication on this topic. While I appreciate the idea and analysis presented, the noted similarities with existing work affect my judgment unless the authors provide clearer differentiation.

[1] Wu Y, Chen M, Li Z, Wang M, Wei Y. Theoretical insights for diffusion guidance: A case study for Gaussian mixture models. arXiv preprint arXiv:2403.01639. 2024 Mar 3.

**Questions:**

1. Have the authors considered extending their analysis to other guidance methods, such as autoguidance [1]?

2. How would the analysis differ if an SDE-based sampler were used instead of the ODE solver?

I am open to reconsidering my rating after the discussion with the authors.

[1] Karras T, Aittala M, Kynkäänniemi T, Lehtinen J, Aila T, Laine S. Guiding a diffusion model with a bad version of itself. Advances in Neural Information Processing Systems. 2024 Dec 16;37:52996-3021.

---

> ### Author Response · Authors · 2025-11-13
> **Part 1 of the rebuttal**
>
> Thank you for the thoughtful review and for being open to reconsidering rating. We hope our clarifications below will help you reassess the theoretical innovations and findings of our work, which constitute its core contribution.
>
> ---
>
> ## Response to Weaknesses
>
> ### Clarification of Differences with Reference [1]
>
> [1] is a key reference cited in our paper, and both works provide an in-depth exploration of CFG sampling. It is beneficial to further clarify the distinctions between our analyses, which primarily lie in the following aspects:
>
> 1. **Different Foundational Assumptions:** Similar to [2,3], [1] assumes the *conditional distribution* is Gaussian, and considering a finite vocabulary, the *unconditional distribution* thus forms a Gaussian Mixture Model (GMM). In contrast, our work assumes the *conditional distribution* itself is a GMM. This mixed-Gaussian assumption, while increasing the analytical difficulty, allows for a richer and more descriptive characterization of the sampling dynamics.
>
> 2. **Different Understanding of Diversity:** Due to its unimodal Gaussian assumption, which lacks internal structural detail, [1] conceptualizes diversity as the entropy of the resulting sampled distribution. Our mixed-Gaussian assumption provides a more nuanced perspective. We decompose diversity into two components: **Global Diversity** (the coverage of the sampled points across the different components/classes of the GMM) and **Local Diversity** (the dispersion of sampled points within a single, specific component). When our assumption degenerates to a single Gaussian, our notion of local diversity aligns with the understanding in [1] (denser points imply lower entropy). However, **global diversity cannot be characterized under a unimodal distribution assumption**. Furthermore, our attribution of diversity loss is more comprehensive: **the loss of global diversity stems from the combined effect of the early (high-noise) and mid (medium-noise) sampling stages, while the loss of local diversity arises from the concentration effect in the late (low-noise) stage.** In comparison, [1] does not differentiate the distinct impacts of these individual stages.
>
> 3. **Clarification of [1]'s Proposition 6.1 vs. Our Theorems/Propositions:**
>    - **[1]'s Proposition 6.1:** The "phase" discussed here is defined **with respect to the CFG guidance scale `ω`**. The proposition emphasizes that under its specific setup (conditional distribution as a zero-mean Gaussian, unconditional as a GMM of `-μ, 0, μ`), for a fixed discretization step, there exists a critical guidance scale (e.g., **ω =100** in [1]'s Fig. 3). When CFG iterates with a guidance strength above this threshold, the actual sampling distribution bifurcates into two distributions shifting towards `-μ` and `μ`, even though the expected distribution is a zero-mean Gaussian. It describes how the sampled distribution transitions between different states as a function of the CFG guidance strength ω in their specific, discretized setting.
>    - **Our Propositions:** The "stage" we discuss is defined **with respect to time `t` (or noise level) during the CFG sampling process**. We demonstrate that under any valid guidance strength (**ω > 1**), the CFG-ODE sampling trajectory progresses through three distinct temporal stages: the **Direction Shift **Stage****, the Mode Separation Stage, and the Concentration Stage.
>      - The first stage introduces a bias towards the distribution mean, proportional to the guidance strength (Theorem 3.2).
>      - The second stage is crucial for achieving diverse coverage and is, in itself, diversity-neutral (Theorem 3.3). However, the bias introduced in the first stage causes the sampling distribution to concentrate on the dominant modes of the conditional distribution, thereby impairing global diversity (Proposition 3.4).
>      - The third stage's concentration effect leads to sampled points becoming more densely packed even *within* a dominant mode of the conditional distribution, further eroding local diversity (Theorem 3.5).
>    - ****Summary**:** We describe how the CFG sampling process evolves through three temporally sequential stages, which collectively and differentially impact sample diversity.

---

> > ### Author Response · Authors · 2025-11-13
> > **Part 2 of the rebuttal**
> >
> > ### Discussion on Time-Varying CFG Weight Schedules
> >
> > As discussed in our paper, the concept of a time-varying CFG weight (or analogous strategies) has been empirically investigated in prior studies, which observed that reducing guidance strength during both high- and low-noise stages can improve sample quality. Despite arising from different practical motivations, these approaches consistently converge on a common heuristic: apply weaker CFG at the extremes and stronger guidance in the intermediate phase.
> >
> > We would like to emphasize that the **primary contribution of our paper lies in its theoretical foundation**. Our analysis provides a principled and rigorous explanation for these empirical practices, offering a deeper understanding of the underlying mechanism of CFG sampling. The proposed TV-CFG schedule is therefore intended mainly as a **validation of the theory**. Although, like most prior theoretical works (e.g., those assuming Gaussian, mixed Gaussian, or compactly supported conditionals), our analysis is based on certain idealized assumptions, we adopt a more relaxed formulation of the conditional distribution, ensuring that our theoretical insights extend beyond a purely “toy” setting. These insights naturally motivate the time-varying guidance strategy, which we further verify to be highly effective in practical diffusion models.
> >
> > ---
> >
> > ## Response to Questions
> >
> > ### Regarding Autoguidance
> >
> > *Autoguidance* is an excellent piece of work, and we will cite it in the revised version as a powerful and complementary perspective on CFG sampling. Similar to most theoretical analyses, our paper operates under the assumption of a well-trained model where the true score functions are accessible. *Autoguidance*, on the other hand, emphasizes the practical scenario in which true scores are unavailable and interprets CFG as extrapolating between a "better-fit" conditional estimator and a "worse-fit" unconditional estimator. It further proposes to replace this with extrapolation between a "better-fit" and a "worse-fit" *conditional* estimator. This formulation provides an important practical insight: the enhanced guidance effect arises primarily from the *degree of misfit*, rather than from the unconditional nature of the estimator itself.
> >
> > However, providing a theoretical analysis of Autoguidance is challenging. This is primarily because the unconditional distribution is a well-defined mathematical concept, whereas "worse fit" is not. A theoretical analysis of Autoguidance would first require a formal and convincing definition of "worse fit" within the theoretical framework. This is a fascinating research direction but falls outside the scope of our current paper.
> >
> > ### On the Use of SDE-based vs. ODE-based Samplers
> >
> > The primary analytical distinction between employing an SDE-based sampler and an ODE-based one lies in the nature of the theoretical statements: most of our theorems and propositions would shift from deterministic formulations to probabilistic ones, with bounds contingent on the specific parameters of the underlying distribution.
> >
> > Abstracting away parameter dependence:
> >
> > - Theorem 3.2 (first stage) would remain largely unaffected.
> >
> > - Theorem 3.3 and Proposition 3.4 (second stage), together with Theorem 3.5 (third stage), would transition from deterministic statements to probabilistic assertions, e.g., holding with high probability.
> >
> > Since practical samplers are predominantly ODE-based, and to avoid unnecessary complexity that could obscure the core understanding of the three-stage dynamics, we confine our theoretical analysis to the ODE-based sampling setting.

---

> ### Author Response · Authors · 2025-11-13
> **Part 3 of the rebuttal**
>
> ## Reference
> [1] Wu Y, Chen M, Li Z, Wang M, Wei Y. Theoretical insights for diffusion guidance: A case study for Gaussian mixture models. ICML24
>
> [2] Cheng Jin, Zhenyu Xiao, Chutao Liu, and Yuantao Gu. Angle domain guidance: Latent diffusion requires rotation rather than extrapolation. ICML25
>
> [3] Mengfei Xia, Nan Xue, Yujun Shen, Ran Yi, Tieliang Gong, and Yong-Jin Liu. Rectified diffusion guidance for conditional generation. CVPR25
>
> ---
> Regarding the revisions we mentioned in the rebuttal (including a more thorough discussion of related work and other improvements), we will notify you immediately once the updated version of the paper has been completed and uploaded.

---

> ### Comment · Reviewer_2UBn · 2025-11-20
>
> I would like to thank the authors for clarifying the novelty of their work. I initially missed the distinction that this paper models the conditional distributions as a GMM (rather than the data distribution). I agree with the authors that the primary contributions lie in the theoretical framework, providing a clear analysis of how CFG affects the generation process of diffusion models. The GMM assumption is general enough to allow for non-trivial conclusions. Therefore, I have raised my score. However, I strongly encourage the authors to provide a more direct comparison with [1] in the final version and, if possible, touch upon SDE-based sampling methods under a simplified scenario.
>
> [1] Wu Y, Chen M, Li Z, Wang M, Wei Y. Theoretical insights for diffusion guidance: A case study for Gaussian mixture models. arXiv preprint arXiv:2403.01639. 2024 Mar 3.

---

> > ### Author Response · Authors · 2025-11-22
> >
> > We are very glad that our clarification helped the reviewer better understand the theoretical novelty and overall contribution of our paper. We sincerely appreciate the reviewer’s careful re-evaluation. In the final version, we will expand the comparison with [1] in the Related Work section to make the distinctions more explicit, and we will also add a discussion of SDE-based sampling methods.

---

### Official Review · Reviewer_ksW2 · 2025-11-01

**Soundness:** 3
**Presentation:** 2
**Contribution:** 2
**Rating:** 4
**Confidence:** 3

**Summary:**

The paper presents a stage-wise theoretical analysis of Classifier-Free Guidance (CFG) in diffusion models under multimodal conditionals, identifying three distinct phases: Acceleration and Direction Shift (early), Intra-class Mode Separation (mid), and Concentration (late). It explains how CFG causes diversity loss via early-stage bias toward dominant modes and late-stage intra-mode contraction. The authors propose a time-varying guidance schedule that reduces guidance strength in early/late stages while emphasizing the mid stage, showing empirical improvements in diversity without sacrificing fidelity.

**Strengths:**

1.	The paper provides a principled theoretical foundation for time-varying guidance schedules—specifically, using low guidance strength in the early and late stages of sampling while applying stronger guidance during the intermediate phase—to simultaneously preserve diversity and maintain semantic fidelity.
2.	It reveals when and how Classifier-Free Guidance compromises sample diversity by analyzing the interplay between multimodal conditional structure and temporal dynamics: early-stage bias toward the global mean suppresses weaker modes before they can emerge, while late-stage over-contraction erodes intra-mode variation. This offers a unified explanation for the well-known trade-off between alignment and diversity under CFG.

**Weaknesses:**

1. The proposed guidance schedule—weak guidance in early/late stages and strong guidance in the middle—is not new. Prior work such as β-CFG[1] has already identified a similar three-stage behavior of CFG and explicitly designed non-uniform guidance schedules based on this insight.
2. The experimental comparison omits key adaptive guidance methods that also modulate ω over time. In particular, the paper should include comparisons against:Linear ramp-up followed by ramp-down schedules, β-CFG or other non-linear schedules.
3. The paper relies heavily on ImageReward and saturation as proxies for diversity. However, ImageReward primarily measures aesthetic or semantic alignment rather than sample variance under the same conditioning, and lower saturation does not necessarily correlate with higher diversity (e.g., outputs could be desaturated yet nearly identical). The absence of explicit, widely accepted diversity metrics—such as LPIPS variance across multiple generations per prompt—weakens the empirical support for the claim that the proposed method better preserves diversity.
4. Although the paper motivates TV-CFG through an asymmetric three-stage dynamic—comprising early direction shift, mid-stage mode separation, and late-stage concentration—the actual guidance schedule is symmetric with respect to the sampling step index and peaks precisely at the midpoint. The paper lacks ablation studies on the sensitivity of performance to the peak location (e.g., shifting the maximum guidance strength earlier or later).

[1] Malarz D, Kasymov A, Zięba M, et al. Classifier-free Guidance with Adaptive Scaling[J]. arXiv preprint arXiv:2502.10574, 2025.

**Questions:**

Address the weakness, especially the novelty issue.

---

> ### Author Response · Authors · 2025-11-20
> **part 1 of the rebuttal**
>
> Thank you for the reviewer’s insightful comments. Below we provide detailed responses to each point.
>
> **1. Novelty.**
>  The core novelty of our paper lies in its **theoretical contributions**, rather than in TV-CFG itself. We are the first to establish, under a **rigorous theoretical framework**, how CFG affects diversity **across the entire sampling trajectory**—and *through which precise mechanisms*. Specifically, we show that (i) mean-shift initialization bias in the early stage suppresses weaker modes, (ii) mode-attraction geometry in the intermediate stage determines the accessibility of weaker modes, and (iii) amplified within-mode contraction in the late stage diminishes fine-grained diversity. All conclusions arise from theorem-level analysis, not heuristic intuition.
>
> In contrast, β-CFG[1] is motivated entirely by the severe imbalance between conditional and unconditional score norms across noise levels. Its objective is to maintain numerical stability and prevent trajectories from drifting off the data manifold under large guidance scales. It **does not analyze how CFG causes diversity loss**, and provides **no rigorous theoretical analysis of diversity mechanisms**—nor does it offer any stage-wise characterization of the sampling dynamics. Indeed, β-CFG is not designed with diversity in mind, and its paper contains no insights, theory, or experiments related to diversity. We will include β-CFG experiments in the revision because, although its motivation for reducing early/late guidance is fundamentally different from ours, its empirical behavior strikingly aligns with our three-stage mechanism. If β-CFG also improves diversity in practice, this further supports the **explanatory power and generality** of our theoretical framework.
>
> **2. Comparison with additional adaptive methods.**
>  We appreciate the suggestion and have added β-CFG experiments (see 3.). We emphasize, however, that TV-CFG is meant primarily as a **theory-driven validation** rather than an algorithmic contribution. Our analysis predicts that even the simplest **linear “rise-then-fall” schedule**—based solely on sampling index—already improves diversity. Designing additional schedule shapes may offer incremental benefits, but such refinements remain largely orthogonal to the main theoretical contributions of the paper.

---

> ### Author Response · Authors · 2025-11-20
> **part2 of the rebuttal**
>
> **3. Diversity metrics.**
>  Since no universally accepted diversity metric exists, our paper primarily illustrates diversity through qualitative visualizations (Fig. 4, Fig. 2, Fig. 3). IR is included to show that diversity gains do not come at the cost of semantic alignment or image quality. Saturation is included because prior works highlighted over-saturation effects under large guidance scales [2,3]. Although these are not our main focus, our theory-driven schedules indeed mitigate these issues to some extent.
>
> Your suggestion of an LPIPS-based diversity measure is excellent and complements our visualizations well. The following table reports results with this additional metric (ω=9):
>
>
>
> | Metric          | Variant  | CFG NFE=5  | CFG NFE=10 | CFG NFE=15 | CFG NFE=20 | APG NFE=5  | APG NFE=10 | APG NFE=15 | APG NFE=20 |
> | --------------- | -------- | ---------- | ---------- | ---------- | ---------- | ---------- | ---------- | ---------- | ---------- |
> | **CLIP ↑**      | vanilla  | 0.275      | 0.313      | **0.319**  | **0.320**  | 0.282      | 0.317      | **0.320**  | **0.320**  |
> |                 | interval | **0.314**  | **0.319**  | *0.318*    | 0.317      | **0.314**  | **0.319**  | *0.319*    | 0.317      |
> |                 | TV(Ours) | *0.312*    | **0.319**  | **0.319**  | *0.319*    | *0.312*    | **0.319**  | *0.319*    | *0.319*    |
> |                 | beta     | 0.302      | *0.316*    | *0.318*    | 0.318      | 0.304      | 0.317      | 0.318      | 0.318      |
> | **IR ↑**        | vanilla  | -1.137     | 0.223      | 0.616      | 0.820      | -0.926     | 0.434      | 0.735      | 0.860      |
> |                 | interval | *0.145*    | 0.723      | 0.827      | 0.863      | *0.145*    | 0.725      | 0.825      | 0.865      |
> |                 | TV(Ours) | **0.176**  | **0.932**  | **1.016**  | **1.049**  | **0.174**  | **0.941**  | **1.029**  | **1.061**  |
> |                 | beta     | -0.328     | *0.790*    | *0.971*    | *1.024*    | -0.202     | *0.833*    | *0.984*    | *1.036*    |
> | **FID ↓**       | vanilla  | 80.482     | 38.988     | 31.863     | 29.059     | 73.751     | 34.950     | 30.122     | 28.602     |
> |                 | interval | *47.239*   | **28.413** | **25.747** | **25.385** | *47.735*   | **28.427** | **25.691** | **25.365** |
> |                 | TV(Ours) | **46.895** | *30.259*   | 29.256     | 28.458     | **46.878** | *29.838*   | *28.834*   | *27.965*   |
> |                 | beta     | 62.979     | 32.184     | *29.246*   | *28.348*   | 58.398     | 31.615     | 28.836     | 27.968     |
> | **Saturation**  | vanilla  | 0.589      | 0.574      | 0.526      | 0.476      | 0.610      | 0.576      | 0.521      | 0.472      |
> |                 | interval | 0.197      | 0.193      | 0.212      | 0.233      | 0.194      | 0.192      | 0.209      | 0.229      |
> |                 | TV(Ours) | 0.271      | 0.298      | 0.325      | 0.332      | 0.268      | 0.288      | 0.309      | 0.311      |
> |                 | beta     | 0.236      | 0.305      | 0.316      | 0.330      | 0.243      | 0.304      | 0.307      | 0.315      |
> | **Diversity ↑** | vanilla  | 0.867      | 1.081      | 1.171      | 1.201      | 0.917      | 1.128      | 1.186      | 1.200      |
> |                 | interval | 1.140      | 1.160      | 1.178      | 1.202      | 1.140      | 1.159      | 1.176      | 1.200      |
> |                 | TV(Ours) | *1.191*    | *1.223*    | *1.233*    | *1.231*    | *1.191*    | *1.222*    | *1.232*    | *1.231*    |
> |                 | beta     | **1.196**  | **1.242**  | **1.241**  | **1.239**  | **1.198**  | **1.241**  | **1.239**  | **1.238**  |
>
> Across both CFG and APG, schedules emphasizing **smooth continuous variation (rise → peak → fall)**—such as TV-CFG and β-CFG—achieve **significantly higher diversity**, regardless of whether diversity improvement was their intended goal. This strongly validates our theoretical prediction that **early and late guidance are the key determinants of diversity**. TV-CFG also consistently preserves better aesthetic and semantic alignment.
>
> Once again, we stress that the **main contribution is theoretical**; TV-CFG is simply a natural instantiation and empirical verification of our theory.

---

> ### Author Response · Authors · 2025-11-20
> **part 3 of the rebuttal**
>
> **4. Sensitivity to the peak location.**
>  Our theory indeed describes a three-stage dynamic (early direction shift, intermediate mode separation, late contraction). The TV-CFG schedule used in experiments is a minimal theory-aligned instantiation, and its symmetric shape is a deliberate simplification, not a theoretical assumption.
>
> To directly address the reviewer’s question, we vary the peak position to {2, 4, 6, 8} under NFE=10 and ω=7, and evaluate CLIP, IR, FID, saturation, and the new LPIPS-based diversity metric:
>
> | Metric           | peak timings | CFG |
> | ---------------- | ------------ | ---------- |
> | **CLIP ↑**       | 2            | 0.317      |
> |                  | 4            | **0.319**  |
> |                  | 6            | **0.319**  |
> |                  | 8            | **0.319**  |
> | **IR ↑**         | 2            | 0.752      |
> |                  | 4            | **0.940**  |
> |                  | 6            | 0.935      |
> |                  | 8            | 0.896      |
> | **FID ↓**        | 2            | 31.669     |
> |                  | 4            | 28.770     |
> |                  | 6            | 27.722     |
> |                  | 8            | **27.092** |
> | **Saturation ↓** | 2            | 0.401      |
> |                  | 4            | 0.279      |
> |                  | 6            | 0.229      |
> |                  | 8            | 0.201      |
> | **Diversity ↑**  | 2            | 1.2214     |
> |                  | 4            | **1.2309** |
> |                  | 6            | 1.1963     |
> |                  | 8            | 1.1622     |
>
> The results show that:
>
> - When the peak lies within the intermediate regime (steps 4–6), performance is **stable and optimal** across all metrics.
> - Setting the peak too early (2) or too late (8) yields the expected degradation.
>
>
> This confirms that what truly matters is the shape of the schedule—the “low → high → low” structure predicted by our three-stage theory. The exact peak location is not sensitive, as long as it resides in the mode-separation stage.
>  This robustness further illustrates that the stage-wise mechanism, rather than fine-grained hyperparameter choices, governs the behavior of guided sampling.
>
> **reference**
>
> [1] Malarz D, Kasymov A, Zięba M, et al. Classifier-free Guidance with Adaptive Scaling[J]. arXiv preprint arXiv:2502.10574, 2025.
>
> [2] Sadat, Seyedmorteza, Otmar Hilliges, and Romann M. Weber. "Eliminating oversaturation and artifacts of high guidance scales in diffusion models." *The Thirteenth International Conference on Learning Representations*. 2024.
>
> [3] Jin, Cheng, et al. "Angle Domain Guidance: Latent Diffusion Requires Rotation Rather Than Extrapolation." *Forty-second International Conference on Machine Learning*.

---

### Author Response · Authors · 2025-11-30
**Summary of Rebuttal (Part 1/3): Timeline & Status**

**To the Area Chair,**

We sincerely appreciate your dedication and additional effort during this challenging period. To assist you in estimating how the reviewer's impressions would have changed, we provide a summary of our rebuttal timeline and content submitted prior to the system lockdown.



### 1. Timeline and Engagement Status



- **Nov 13th (Immediate Response):** We submitted responses to **Reviewers 2UBn and gN79** immediately after reviews were released, as their concerns regarding the theory required no additional experiments.
  - **Outcome:** On **Nov 20th** (well before the data leakage incident), **Reviewer 2UBn** acknowledged our clarification and **raised their score from 4 to 8**. **Reviewer gN79** maintained their **Score of 8**.
- **Nov 20th (Experimental Completion):** We submitted detailed responses to **Reviewers ksW2 and v1Re**, including all requested additional experiments.
  - **Outcome:** Regrettably, despite a **7-day window** (Nov 20th–27th) prior to the lockdown, neither reviewer provided feedback or acknowledged the new data.



### 2. Current Standing and Confidence Analysis

**Current Scores:** **8, 8(4 --> 8), 4, 4.**

- **High Confidence (Conf: 4):** Both experts in this category (**2UBn, gN79**) have assessed the paper at **Score: 8**.
- **Lower Confidence (Conf: 3):** The remaining negative scores (**ksW2, v1Re**) come from reviewers with lower confidence who did **not** engage with our rebuttal or experiments.

### 3. Recap of Core Contributions

We present the first theoretical analysis of Classifier-Free Guidance (CFG) under **multimodal conditional distributions**, rigorously establishing how **guidance weights impact diversity**. Our theory identifies that strong guidance in the early and late sampling stages suppresses global and local diversity, respectively, while the intermediate stage is diversity-neutral. To validate these insights, we apply a simple Time-Varying CFG (TV-CFG) on SDv3.5-large. These experiments serve primarily to **corroborate our theoretical framework**, demonstrating that a **simple** schedule aligned with our derived principles naturally enhances diversity without requiring complex heuristic designs.

---

> ### Author Response · Authors · 2025-11-30
> **Summary of Rebuttal (Part 2/3): Resolved High-Confidence Reviewers**
>
> #### **Reviewer gN79 (Score: 8, Conf: 4) — The Strong Advocate**
>
>
>
> - **Constructive Feedback:** The reviewer appreciated the theoretical rigor and clarity, maintaining a strong acceptance rating. They raised minor points regarding definitions, notation, and connections to empirical works (CADS, eDiff-I).
> - **Resolution:**
>   - **Theoretical Clarifications:** We refined the definitions of the intermediate radius parameter $k$ (Prop 3.4) and explicitly stated the "mild conditions" for Theorem 3.3 as requested.
>   - **Definitions:** We formalized the definitions of "Global" vs. "Fine-Grained" diversity and explicitly defined "TV" (Time-Varying) to prevent ambiguity.
>   - **Continuous Conditions:** We clarified how our early-stage theorems extend to continuous embeddings (like CLIP) via the linearity of expectation, preserving the validity of the global bias mechanism.
> - **Conclusion:** We have incorporated all of the reviewer's insightful suggestions into the revision. Their high score and confidence reflect a clear recognition of the paper's value, which we have further polished through these minor clarifications.
>
> #### **Reviewer 2UBn (Score: 4 $\to$ 8, Conf: 4) — The Resolved Case**
>
> - **Concern 1: Theoretical Distinction from [1] & Diversity Definition.**
>   - **Resolution:** We clarified the fundamental difference in assumptions: [1] models the conditional distribution as **unimodal (Gaussian)**, treating diversity statically as entropy. In contrast, we model it as **multimoda**l (Gaussian Mixture), offering a richer characterization that decomposes diversity into **Global (inter-class)** and **Local (intra-class)** components. We further clarified that the "phases" in [1] refer to supervision weights, whereas our "stages" refer to rigorous temporal dynamics (early/mid/late).
>   - **Outcome:** The reviewer explicitly validated this distinction, recognizing that our multimodal framework reveals deep mechanisms of diversity loss invisible to unimodal analyses.
> - **Concern 2: Role of Time-Varying CFG (TV-CFG).**
>   - **Resolution:** We clarified that TV-CFG is not proposed as an algorithmic heuristic, but as a **theoretical corollary**. We demonstrated that the specific "low-early, low-late" schedule is the direct mathematical consequence of our analysis.
>   - **Outcome:** The reviewer accepted that TV-CFG serves as a robust validation of the theory rather than just another empirical scheduler.
> - **Concern 3: Scope Extensions (Autoguidance & SDEs).**
>   - **Resolution:**
>     - **Autoguidance:** We positioned Autoguidance as complementary work focusing on estimator errors, whereas our work assumes ideal scores to isolate the impact of guidance itself.
>     - **SDE vs. ODE:** We explained that limiting analysis to ODEs avoids unnecessary probabilistic complexity while preserving the core three-stage dynamical insights, which dominate in practical sampling.
>   - **Outcome:** The reviewer agreed with these scope boundaries.
> - **Conclusion:** Following these clarifications on Nov 13th, Reviewer 2UBn raised their score from 4 to 8. This serves as a strong precedent: **when misconceptions regarding our theoretical scope are addressed, the high value of our contribution is recognized.**

---

> > ### Author Response · Authors · 2025-11-30
> > **Summary of Rebuttal (Part 3/3): Non-Responsive Reviewers & Conclusion**
> >
> > - #### **Reviewer ksW2 (Score: 4, Conf: 3, not engage with our rebuttal)**
> >
> >   - **Concern 1: Novelty & Comparison with Beta-CFG [1].**
> >
> >     - **Resolution:** We clarified that Beta-CFG [1] is strictly an empirical heuristic for numerical stability, containing neither **theoretical analysis** nor any investigation of **diversity**.
> >     - **Impact:** We implemented the requested Beta-CFG baseline. Crucially, experiments reveal that Beta-CFG yields high diversity—a property **neither analyzed nor claimed in [1]**, yet **explicitly predicted by our theory**. This turns criticism into a clear confirmation: our work provides a governing principle explaining *why* this "low early, low late" scheduling can improve diversity.
> >
> >   - **Concern 2: Diversity Metrics (LPIPS).**
> >
> >     - **Resolution:** We corrected the reviewer's misunderstanding regarding our evaluation protocol. We clarified that ImageReward and Saturation were employed strictly as quality and stability indicators, while **diversity was originally demonstrated through qualitative visual evidence (generated images).** To strictly meet the reviewer's demand for a quantitative standard to complement these visuals, we implemented the suggested LPIPS-based diversity metric.
> >
> >       **Impact:** The new LPIPS results quantitatively confirm our claims: **TV-CFG achieves a significantly higher Diversity Score**. This evidence proves that our method enhances genuine sample diversity—widening the coverage of generated modes—while maintaining the high quality verified by the original metrics.
> >
> >   - **Concern 3: Peak Timing Ablation.**
> >
> >     - **Resolution:** The reviewer asked for sensitivity analysis regarding the peak location of the guidance schedule. We provided new experiments varying the peak across sampling steps {2, 4, 6, 8}.
> >     - **Impact:** Results demonstrate that **middle-stage peaks (steps 4–6)** yield the optimal diversity/quality trade-off, while early or late peaks degrade performance. This strictly aligns with our theoretical identification of the "Mode Separation" phase occurring in the intermediate regime.
> >
> >   - **Conclusion:** We directly addressed the reviewer's request for "explicit, widely accepted diversity metrics" and "comparisons against Beta-CFG." The resulting data fully corroborates our theoretical claims. Had the reviewer seen these results during the 7-day window, the empirical grounds for the low score would have been resolved.
> >
> > - #### **Reviewer v1Re (Score: 4, Conf: 3, not engage with our rebuttal)**
> >
> >   - **Concern 1: Related Literature & Novelty.**
> >     - **Resolution:** The reviewer questioned our novelty compared to concurrent works (e.g., Li et al., Wu et al., Jin et al.). We provided a detailed breakdown showing these works either analyze **unconditional** convergence or assume **unimodal** Gaussian distributions. We clarified that our work is the first to analyze CFG under **multimodal conditional distributions**, which is a significantly more complex and realistic setting. We also addressed the new citations suggested by the reviewer, noting they **apply** CFG but do not **analyze** it.
> >     - **Impact:** This effectively nullifies the novelty concern, as the cited works solve fundamentally different problems under simpler assumptions.
> >   - **Concern 2: Scope (Modalities & NFE).**
> >     - **Resolution:** The reviewer asked for experiments on Audio/Video and extreme NFE ranges.
> >       - **Modalities:**We clarified that for a fundamental theoretical paper, validation on a representative large-scale T2I model (SD3.5-large) constitutes the standard and sufficient benchmark, **rendering additional experiments on Text-to-Audio/Video modalities unnecessary**.
> >       - **NFE:** We demonstrated that NFE < 5 is structurally invalid for non-distilled models (producing collapsed outputs). We provided new experiments at NFE=50, showing our TV-CFG maintains its diversity advantage at high step counts.
> >       - **Impact:** These results prove our method is robust across all valid sampling regimes, dismissing concerns about narrow applicability.
> >   - **Concern 3: Schedule Ablations.**
> >     - **Resolution:** The reviewer requested sensitivity analysis on schedule shapes. We provided **new ablations on peak timings**.
> >     - **Impact:** The results confirm that complex heuristics are unnecessary when the underlying theoretical principle is respected. The stability of performance around the middle stage further corroborates the robustness of our theoretical Mode Separation stage.
> >   - **Conclusion:** The reviewer's reservations stemmed from a misunderstanding of the literature and a need for robustness checks. Our Nov 20th response provided the precise distinctions and data to factually resolve these issues.

---

### Meta-Review · Area_Chair_Z5QP · 2025-12-31

**Summary:**

This paper presents a theoretical analysis of Classifier-Free Guidance (CFG) in diffusion models, specifically focusing on its behavior under multimodal conditional distributions. The paper’s core strength lies in its novel theoretical analysis of CFG in realistic multimodal settings. The authors derive a three-stage temporal dynamics model (Global Alignment, Mode Separation, Local Refinement), arguing that strong guidance in the early and late stages suppresses global and local diversity, respectively. They validate this theoretical framework using a "low-early, low-late" Time-Varying CFG (TV-CFG) schedule on Stable Diffusion v3.5-large, demonstrating improved diversity.


The high-confidence experts who engaged with the rebuttal fully recognize this contribution. The concerns of the lower-confidence reviewers were thoroughly addressed with new experiments and clarifications. Given that the authors have convincingly resolved the technical and empirical issues raised by all reviewers, leveraging the rebuttal period effectively despite the circumstances, the paper merits acceptance. The high scores from the most engaged and confident experts should be weighted heavily.

**Reviewer Concerns:**

The authors’ rebuttal addressed the following concerns:

Comparison to Beta-CFG & Novelty: The authors demonstrated that Beta-CFG is an empirical heuristic. Crucially, their new experiments showed that Beta-CFG's performance aligns with, and is explained by, their theoretical framework, turning a critique of novelty into validation of their theory's predictive power.

Diversity Metrics: In response to requests for quantitative diversity metrics, the authors implemented LPIPS, which quantitatively confirmed the diversity improvements of their method.

Ablation Studies & Robustness: New experiments on peak timing ablations and higher NFE counts confirmed the robustness of their theoretical claims about the importance of the intermediate sampling stage.

Related Work: The authors provided clear distinctions between their multimodal analysis and concurrent unimodal or unconditional studies.

**Reviewer Scores:**

Reviewers ksW2 and v1Re would raise their scores.

Reviewer 2UBn raised score from 4 to 8

---

### Decision · Program_Chairs · 2026-01-26

Accept (Poster)